# FT-Dojo: Towards Autonomous LLM Fine-Tuning with Language Agents

**Qizheng Li** [1][2][*]  **Yifei Zhang** [1][3][*]  **Xiao Yang** [1][*]  **Xu Yang** [1][*]
**Zhuo Wang** [1][4]  **Bowen Xian** [1]  **Weiqing Liu** [1][†]  **Jiang Bian** [1]

## Abstract

Fine-tuning large language models for vertical domains remains labor-intensive, requiring practitioners to curate data, configure training, and iteratively diagnose model behavior. Despite growing interest in autonomous machine learning and language agents, end-to-end LLM fine-tuning has not been systematically studied as an interactive agent task. We introduce FT-Dojo, an interactive benchmark environment for autonomous LLM fine-tuning, comprising 13 tasks across 5 domains. Rather than a new collection of static datasets, FT-Dojo standardizes a task interface, shared raw-data repository, sandboxed execution environment, structured feedback protocol, and held-out evaluation procedure. We further develop FT-Agent, a fine-tuning-oriented autonomous framework that uses structured iteration planning, fail-fast validation, and multi-level feedback analysis to refine data and training strategies. Experiments show that FT-Agent provides a strong initial baseline, achieving the best performance on 10 out of 13 tasks, with additional controlled comparisons against frontier agents, open-source planning backbones, and multi-run statistics supporting the main findings. Case studies show that agents can recover from failures through cumulative learning, while still exposing limitations in causal diagnosis and long-horizon planning. The implementation is available at https://github.com/microsoft/rd-agent.

## 1. Introduction

Large language models have achieved remarkable general capabilities (Liu et al., 2025; Team et al., 2025), yet deploying them in specialized domains such as chemistry, le-

gal, and finance often demands domain-specific adaptation. While prompt engineering offers a lightweight entry point, production-grade applications typically require fine-tuning to meet reliability and performance standards (Zhang et al., 2023; Huang et al., 2024). However, fine-tuning remains a labor-intensive process that must be re-engineered for each new domain: domain experts and ML engineers must collaborate to interpret requirements, curate unstructured data, configure training pipelines, and rigorously evaluate model checkpoints. A natural next frontier emerges: **can agents automate the end-to-end LLM fine-tuning process?**

Recent advances in AI-for-AI illustrate a steady broadening of agent capabilities, paving the way to automate the labor-intensive work involved in AI research and development. Benchmarks such as MLE-Bench (Chan et al., 2024) and DSBench (Jing et al., 2024) evaluate agents on tasks involving data preprocessing, feature engineering, and model training, quantitatively measuring how well an agent automates the end-to-end Machine Learning Engineering (MLE) process. However, end-to-end LLM fine-tuning is substantially more open-ended and complex, compared with traditional MLE: (i) Data must be constructed, not merely processed: practitioners navigate heterogeneous raw sources, leveraging diverse tools such as LLM APIs for synthesis and domain-specific processors for cleaning; (ii) Analysis signals extend beyond aggregate metrics: practitioners interpret training dynamics, validation curves, and sampled model outputs to diagnose capability gaps. The iterative joint optimization over data construction and LLM finetuning over various analysis signals has neither been formally defined as an agent task, nor systematically evaluated.

Motivated by these challenges, we introduce **FT-Dojo**, the first interactive benchmark environment for evaluating agents on LLM fine-tuning. The open-ended, iterative refinement inherent to agentic LLM finetuning necessitates a sandboxed interactive environment in which agents can repeatedly refine their approach based on evaluation feedback. To approximate real-world end-to-end LLM finetuning, FT-Dojo aggregates diverse raw data sources into a shared repository: agents must autonomously identify relevant sources, decide how to filter and transform them into training instances, and leverage LLM APIs and other tools for data synthesis and cleaning. The contribution is there-

---

[*]Equal contribution [1]Microsoft Research Asia [2]Peking University [3]Nanjing University [4]The University of Chicago. Correspondence to: Weiqing Liu <Weiqing.Liu@microsoft.com>.

*Proceedings of the 43rd International Conference on Machine Learning*, Seoul, South Korea. PMLR 306, 2026. Copyright 2026 by the author(s).

fore not a new collection of static datasets; rather, FT-Dojo standardizes an interactive protocol, execution environment, feedback interface, and held-out evaluation procedure for autonomous fine-tuning. This design enables both data construction and finetuning configuration to be first-class optimization targets. FT-Dojo comprises 13 tasks spanning 5 domains (Math, Patent Examination, Chemistry, Finance, and Table QA).

In FT-Dojo, we observe that general-purpose agents such as OpenHands (Wang et al., 2024) perform poorly even when supported by strong LLM backends and the same fine-tuning tools. Our analysis shows that general-purpose agents often overlook key signals in noisy feedback, leading to low-value refinements and significant resource waste—an especially acute issue in LLM-finetuning scenarios. Motivated by this gap, we design **FT-Agent** to better align with the fine-tuning scenario. FT-Agent draws on expert practice and introduces targeted mechanisms to address the main obstacles we observe in FT-Dojo: (i) *structured iteration planning* that keeps growing context manageable and helps the agent focus on high-level hypotheses instead of verbose logs, (ii) *fail-fast validation* that prevents wasted computation by catching configuration and data problems early, and (iii) *structured feedback analysis* that helps the agent interpret metrics, loss curves, and error cases to decide how to refine data and training strategies. Through this design, FT-Agent delivers stable, efficient, and meaningful analysis signals across iterations in FT-Dojo, enabling the backend LLM to generate deeper and more insightful refinement proposals.

Our contributions are threefold: (i) We introduce **FT-Dojo**, the first interactive benchmark environment for end-to-end LLM fine-tuning, where agents must autonomously navigate from raw data sources to trained models across 13 tasks in 5 domains, with both data strategy and training configuration as optimization targets; (ii) We propose **FT-Agent**, a fine-tuning-oriented agent framework that introduces three targeted mechanisms described above. These mechanisms help the agent avoid basic engineering pitfalls, focus on high-level decisions, and better reveal algorithmic capabilities of the underlying LLM, providing a strong initial baseline for the benchmark. (iii) Experiments show FT-Agent achieves the best performance on 10 of 13 tasks, and additional frontier-agent baselines, open-source backbone variants, and 3-run statistics support the main qualitative findings. Case analyses reveal emergent capabilities (failure recovery through historical memory, iterative self-improvement) and fundamental limitations in causal reasoning, highlighting the promise and boundaries of autonomous LLM fine-tuning.

**Conflict of Interest Disclosure.** Several authors are employed by Microsoft Research Asia, which developed the FT-Agent system and the R&D-Agent codebase evaluated and released in this paper. The experiments also compare against third-party frontier agent systems and planning backbones under the controlled FT-Dojo protocol.

## 2. FT-Dojo

### 2.1. Task Definition

Given a task specification $\tau$ describing the target capability, a data repository $\mathcal{D}$ containing heterogeneous raw data sources, and an evaluation function $\mathcal{F}$, an agent must autonomously produce a fine-tuned model $M$ by: (1) selecting and processing a subset $d \subseteq \mathcal{D}$ into training instances, (2) configuring training hyperparameters $\theta_t$, and (3) iteratively refining based on evaluation feedback $f^{(t)}$.

Formally, let $\theta_d$ denote the data strategy (source selection, filtering, formatting), $\theta_t$ the training hyperparameters, and $f^{(t)} = \mathcal{F}_\tau(M^{(t)})$ the structured feedback returned by the evaluator at iteration $t$ (with $f^{(0)} = \varnothing$ for the initial iteration). The agent's objective is:

$$\theta_d^*, \theta_t^* = \arg\max_{\theta_d, \theta_t} \mathcal{F}_\tau\big(\texttt{Train}(\theta_d(\mathcal{D}), \theta_t; f^{(t-1)})\big) \quad (1)$$

where $\mathcal{F}_\tau$ is the task-specific evaluator that assesses model performance on the held-out benchmark associated with task specification $\tau$ (accessed via the Meta API $\mathcal{I}$), $\theta_d(\mathcal{D})$ denotes the training data obtained by applying strategy $\theta_d$ to the data repository $\mathcal{D}$, and $\texttt{Train}(\cdot; f^{(t-1)})$ executes fine-tuning informed by previous feedback and returns the resulting model.

### 2.2. System Architecture

FT-Dojo adopts a modular architecture where components operate independently through well-defined interfaces, as illustrated in Figure 1. This design enables flexible task extension and ensures reproducible evaluation.

Each task executes within a sandboxed Docker container that ensures reproducibility and security. The sandbox provides: (1) workspace isolation with separated private storage; (2) configurable resource constraints including GPU/CPU limits and execution timeouts; and (3) secure access to external LLM APIs for data synthesis. Within this environment, agents interact with three core interfaces:

**Meta API ($\mathcal{I}$).** Exposes task specifications, system capabilities, data descriptions, and API documentation through a unified interface, enabling programmatic access to task requirements, available resources, and permitted operations.

**Data Repository ($\mathcal{D}$).** A shared pool of heterogeneous data sources aggregated across all benchmark tasks. Unlike prior benchmarks that provide fixed datasets per task, FT-Dojo creates an open decision space where data selec-

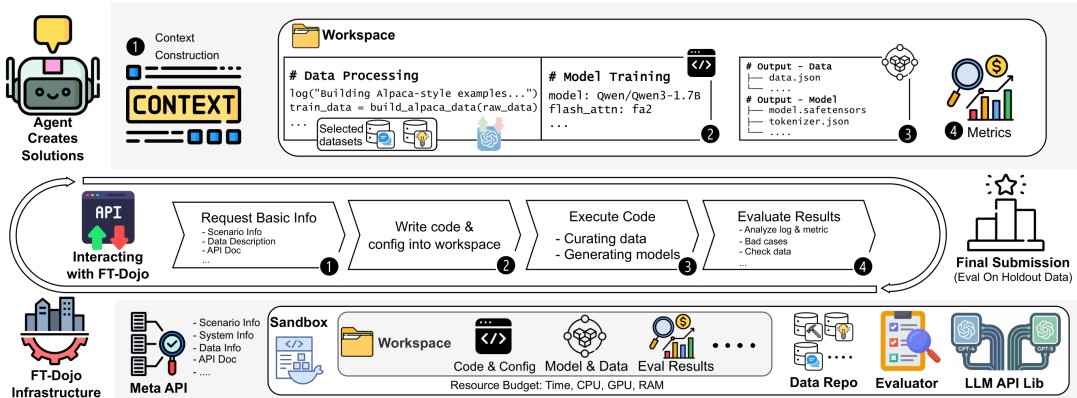

*Figure 1.* Overview of FT-Dojo architecture and agent interaction workflow. Agents operate within an isolated sandbox and interact with three core interfaces: Meta API $\mathcal{I}$ for task and system information, Data Repository $\mathcal{D}$ for training data, and Evaluator $\mathcal{F}$ that returns structured feedback. Agents iteratively query information, submit code and configurations, execute training, and refine based on evaluation feedback until achieving satisfactory performance.

tion, filtering, and processing become integral parts of the optimization problem.

**Evaluator ($\mathcal{F}$).** Adapted from OpenCompass (Contributors, 2023), the evaluator assesses submitted models on held-out data using task-specific metrics. It returns structured feedback comprising: (1) aggregate metric scores, (2) per-instance predictions with correctness labels, and (3) training loss trajectories, enabling fine-grained diagnosis of failure modes.

### 2.3. Environment Formulation

FT-Dojo models autonomous LLM fine-tuning as an iterative agent-environment interaction. At each iteration, the agent observes the task specification $\tau$, data repository $\mathcal{D}$, and evaluation feedback from prior submissions, then selects actions from four core operations: querying task and resource metadata, processing raw data into training instances, executing fine-tuning pipelines, and submitting models for evaluation. After each submission, structured feedback from the evaluator $\mathcal{F}$ enables the agent to diagnose failures, adjust its strategy, and iterate.

### 2.4. Task Suite and Evaluation

Each task in FT-Dojo is specified through two components: (1) a **task objective**, which is a natural language description specifying the domain, target capability, and expected output format; and (2) an **evaluation metric**, which is a script that measures model performance on held-out instances using domain-appropriate metrics.

**Task Suite.** FT-Dojo comprises 13 tasks spanning five domains, as summarized in Table 1. These domains were selected to cover diverse fine-tuning challenges: scientific

reasoning (Math, Chemistry), domain-specific language understanding (Patent Examination, Finance), and structured data comprehension (Table QA). Training data for each domain is curated from diverse sources, including domain-specific corpora and benchmark-associated datasets. Evaluation sets are drawn from established benchmarks, with strict separation from training data (details in Appendix A).

**Benchmark Framing.** FT-Dojo is not intended as a claim that all constituent static datasets are newly introduced. The benchmarked object is the autonomous fine-tuning process under a unified interactive protocol. FT-Dojo standardizes the task interface, shared raw data repository, sandboxed execution environment, validation/test evaluation protocol, and structured feedback returned after each iteration. Existing benchmarks serve as held-out evaluators for domain-specific target capabilities, while FT-Dojo evaluates whether an agent can construct data, configure training, run fine-tuning, and improve from feedback.

**Evaluation Protocol.** FT-Dojo employs a two-phase evaluation protocol. After context-length filtering, each benchmark or subtask is partitioned into non-overlapping validation and test splits with $N_{val} = N_{test} = \min(100, \lfloor n/2 \rfloor)$, where $n$ is the number of filtered examples. Before *final assessment*, agents can submit models for validation-set evaluation, receiving structured feedback to support iterative refinement. The test set remains inaccessible during optimization; for *final assessment*, agents submit their best validation-selected model for held-out test evaluation. The training-data cap is enforced as a shared resource budget: each method may select, filter, or synthesize up to the allowed number of training instances using its own procedure, rather than training from a single precomputed subset.

**Extensibility.** To support flexible extension, both task ob-

*Table 1.* Task suite in FT-Dojo spanning 13 tasks across 5 domains. Data Quality column describes training data characteristics relevant for fine-tuning. Valid/Test indicates the size of both validation and test sets (identical for each task). See Appendix A for metric definitions and abbreviations.

| Domain | Task | Description | Train Data Quality | Metrics | Train | Valid/Test |
|---|---|---|---|---|---|---|
| Mathematics | AIME 2025 | Competition-level mathematical reasoning requiring multi-step symbolic derivations, algebraic manipulation, and exact numerical answers | 82% lack solution No CoT | Accuracy | 40315 | 15 |
| Patent Examination | Prior Art Retrieval | Given a patent claim and candidate patents, identify which were cited as prior art for rejecting the claim | No CoT Q&A pairs only High complexity | Acc, Macro-F1 | 54028 | 100 |
| | Novelty Classification | Classify patent claims as allowed, §102 rejection (lacks novelty), or §103 rejection (obvious) | | Acc, Macro-F1 | 136211 | 100 |
| | Paragraph Identification | Identify the most relevant prior art paragraph from candidates that supports claim rejection | | Accuracy | 64210 | 100 |
| Chemistry | Molecular Understanding | Predict molecular properties from SMILES: functional group counting, scaffold extraction, equivalence | Structured CoT Expert validated | MAE, Tanimoto | 6319 | 160 |
| | Molecule Editing | Modify molecular structures by adding, deleting, or substituting functional groups | | Acc, Tanimoto | 4497 | 50 |
| | Molecule Optimization | Optimize molecules for target properties (binding affinity, solubility) while preserving scaffolds | | SR, Valid Ratio | 5587 | 300 |
| | Reaction Prediction | Predict reaction products, recommend conditions, and select plausible reaction mechanisms | | FTS, BLEU | 6820 | 237 |
| Finance | Financial QA | Answer Chinese financial certification exam questions requiring quantitative analysis and domain reasoning | No CoT MCQ format | Accuracy | 6179 | 100 |
| Table QA | Data Analysis | Perform correlation analysis, trend forecasting, and statistical reasoning over tabular data | With CoT Text & code reasoning | Accuracy | 7372 | 170 |
| | Fact Checking | Verify factual statements against tabular evidence through value lookups and boolean verification | | Exact Match | 2282 | 48 |
| | Numerical Reasoning | Perform arithmetic operations (sum, average, percentage, comparison) over table data | | Accuracy | 8892 | 197 |
| | Visualization | Generate Python code to create specified charts and visualizations from table data | | Pass@1, ECR | 1115 | 25 |

jectives and evaluation metrics are designed as pluggable modules with unified interfaces. Task objectives are registered through the Meta API $\mathcal{I}$, while evaluation metrics are implemented as standalone scripts following a standardized protocol: each script receives model predictions and ground-truth labels as input, and returns structured feedback through the Evaluator $\mathcal{F}$. Despite varying metrics across domains (e.g., accuracy for classification, ROUGE for generation, Tanimoto similarity for molecular tasks), this unified interface enables users to extend FT-Dojo with custom tasks by simply providing a task description and evaluation script, without modifying the core infrastructure.

## 3. FT-Agent

### 3.1. Challenges in Autonomous LLM Fine-Tuning

There are currently no open-source agents designed specifically for large-scale model post-training. Existing systems such as OpenHands (Wang et al., 2024) are built for general-purpose code execution and automation, rather than the high-complexity decision-making and multimodal feedback interpretation required for end-to-end fine-tuning. In FT-Dojo, we thoroughly evaluated general-purpose agents such as OpenHands and found that they still perform poorly even when powered by the strongest LLM backends (Table 3). In some tasks, their performance even falls below that of the original model without any fine-tuning. These systematic observations suggest that general-purpose agents are not well aligned with the following three challenges in autonomous LLM fine-tuning, which collectively undermine the feasibility of autonomous LLM fine-tuning.

**Challenge 1: Long and ever-growing context overwhelms the agent and hides future effective directions.** Fine-tuning produces a large and varied stream of artifacts, including verbose training logs, model checkpoints, loss curves, and evaluation results. With multiple rounds of

iteration, these artifacts accumulate quickly, and the context grows to a scale that becomes difficult for an agent to manage. This hinders the agent from making meaningful decisions for future iterations.

**Challenge 2: Inefficient Exploration.** End-to-end fine-tuning is computationally expensive, making efficient iteration essential. Existing agents, however, lack such fail-fast capabilities. They tend to execute full-scale training even when the configuration is malformed, data processing scripts are buggy, or the dataset is incompatible with the task format. This leads to wasted computation, significantly fewer effective iterations, and poor overall performance in FT-Dojo.

**Challenge 3: Limited Understanding of Evaluation Feedback.** Evaluation in fine-tuning is multifaceted: aggregate metrics show overall ability, per-instance outputs expose consistent reasoning errors, and loss curves reflect optimization behavior. Generic agents often fail to learn from this mixed and unstructured feedback, causing them to miss key diagnostic signals and gain too little useful information even after costly iterations.

To obtain more meaningful benchmark results, we aim to evaluate LLMs at the algorithmic level instead of letting them get stuck on engineering issues. We introduce an agent framework called FT-Agent, which draws on expert practice and provides core mechanisms that help better assess the backend LLM.

### 3.2. FT-Agent Framework

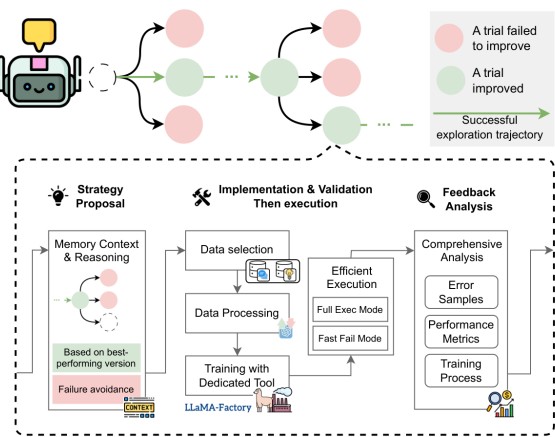

*Figure 2.* Overview of FT-Agent. The agent iteratively explores configurations, learning from failed trials (red) to reach successful ones (green). Each iteration follows three stages: Strategy Proposal, Implementation & Validation, and Feedback Analysis.

FT-Agent operates as a three-stage loop executed at every iteration, where each stage directly addresses one challenge outlined above.

**Stage 1: Strategy Proposal.** Given the task specification retrieved through the Meta API $\mathcal{I}(\tau)$, distilled summaries of prior iterations, structured evaluation signals, and the current system state, the agent formulates a unified hypothesis (see Appendix G.1) that jointly covers high-level data strategy and high-level training configuration. Each iteration produces a compact plan with three parts: (i) a data strategy describing source selection, filtering rules, formatting choices, or synthetic augmentation; (ii) a training configuration covering learning-rate settings, batch-size scale, prompt-format adjustments, CoT usage, or iteration length; and (iii) a rationale that links the proposed changes to observed failure patterns.

This unified hypothesis is generated through an experience-driven process: the agent inherits the current best-performance configuration as the baseline, reviews sibling attempts to understand what has already been explored, and consults past failures to avoid repeating ineffective solutions. Formally, given the best configuration $(\theta_d^*, \theta_t^*)$ and experience $\mathcal{E}$,

$$(\theta_d^{(t+1)}, \theta_t^{(t+1)}) = \texttt{Propose}\big(\tau, (\theta_d^*, \theta_t^*), \mathcal{E}\big) \quad (2)$$

By enforcing high-level descriptions instead of accumulating low-level operational details, FT-Agent keeps iteration histories manageable. These structured summaries support both forward planning and backward review, helping the agent focus on conceptual iteration hypotheses without being overwhelmed by growing fine-grained logs.

**Stage 2: Implementation & Fail-Fast Validation.** To address the inefficiencies of generic agents, FT-Agent applies progressive validation through three increasingly costly stages before full training is allowed to run: (i) static and schema validation, which performs syntax checking, dependency analysis, path verification, and detection of configuration errors; (ii) data format checks and mini-run validation, which run on a small sample or reduced-epoch test to verify output structure and catch broken data pipelines, unstable losses, or incompatible formats; and (iii) runtime sanity detection, which identifies anomalies such as exploding loss, empty datasets, or invalid gradients during a short mini-batch run.

Configurations that fail any stage are rejected immediately, and the agent receives targeted diagnostic logs that encourage rapid correction. This fail-fast process catches most errors before significant computation is spent, enabling much higher iteration throughput. As seen in §4.5, FT-Agent completes 7 loops within 10 hours on Fact_Checking, while OpenHands issues 410 conversation events in a single iteration before timing out. Once all validations pass, the full pipeline runs, including complete data curation $\theta_d(\mathcal{D})$, model fine-tuning $\texttt{Train}(\cdot)$, and evaluation on the validation set.

**Stage 3: Structured Feedback Aggregation and Expert-Guided Diagnosis.** After a successful training run, the evaluator returns multi-level structured feedback, including: (i) overall domain metrics, (ii) per-instance predictions with error tags, (iii) loss curves showing training dynamics, and (iv) sampled failure cases that reveal recurring error patterns.

FT-Agent analyzes these signals to identify capability gaps, signs of overfitting or underfitting, domain mismatch, noise sensitivity, and systematic reasoning problems such as missing CoT structure, table parsing mistakes, or molecule-level errors. This diagnosis supports coordinated refinement: the agent can expand data to cover failed patterns while also adjusting training settings such as learning rate or training duration. If the current run outperforms the previous best, FT-Agent updates the best configuration,

$$(\theta_d^*, \theta_t^*) \leftarrow (\theta_d, \theta_t) \quad \text{if} \quad \mathcal{F}_\tau(M) > \mathcal{F}_\tau(M^*)$$

The full experiment and its feedback are recorded as experience for later iterations. Through this feedback-driven loop, the agent forms clear hypotheses about what failed and how to improve next, enabling steady, expert-like progress across iterations.

## 4. Experiments

### 4.1. Experimental Setup

We evaluate on the 13 tasks across 5 domains in the FT-Dojo suite (Table 1). Each experiment is conducted on a single NVIDIA B200 GPU under a strict 12-hour end-to-end wall-clock budget, which includes agent interaction, data cleaning when applicable, fine-tuning, validation, and final evaluation. Qwen2.5-7B-Instruct (Qwen et al., 2025) serves as the target base model. To simulate data-scarce scenarios and ensure fair comparison, we restrict training to standard Supervised Fine-Tuning (SFT) or LoRA, capped at a maximum of 2,000 training samples per task. This cap is a per-method resource budget: each method independently selects, filters, or synthesizes its own training instances under the same limit. To account for variance in the agent's stochastic optimization process, we run FT-Agent three times with independent initializations and report mean ± standard deviation across runs. Running 3 seeds for all methods and all 13 tasks would require over 1,500 GPU-hours, so we report multi-run statistics for FT-Agent and single controlled runs for the comparison baselines.

**Controlled Protocol.** Across autonomous-agent systems, we hold fixed the external resources and runtime conditions: the same target base model, raw data repository, Docker sandbox, LlamaFactory/OpenCompass wrappers, LLM API access, GPU hardware, validation/test split, and 12-hour wall-clock budget. What varies is the agent framework: FT-Agent uses fine-tuning-oriented orchestration, while Open-Hands retains a general-purpose CodeAct-style control policy under the same external tools. This factorization is intended to isolate the effect of fine-tuning-oriented agent design from differences in tool availability or runtime conditions.

**Model Configuration.** FT-Agent utilizes GPT-5.2 as the interactive backbone. For data processing, we employ GPT-5 and GPT-4o-mini specifically for data cleaning and synthesis. For evaluations that require an LLM to judge correctness against the ground truth, we use GPT-5 as the judge model with a fixed rubric and identical prompts across all methods (Appendix A.4).

**Baselines.** We compare FT-Agent against three distinct paradigms: (1) **Base Model**: The un-tuned `Qwen2.5-7B-Instruct` model. (2) **Manual SFT**: A constrained, resource-matched manual baseline executed by senior NLP practitioners using rule-based cleaning and manual tuning without LLM assistance. This baseline is not intended as an expert upper bound; Appendix B describes its two-pass constraint and an LLM-assisted variant. (3) **Tool-Augmented OpenHands**: A strong baseline based on OpenHands (Wang et al., 2024). Crucially, we equipped this agent with the identical fine-tuning primitives (LlamaFactory wrappers) and runtime environment as FT-Agent. (detailed description in Appendix B).

### 4.2. Main Results

Table 2 presents the comparative evaluation of FT-Agent against three baselines across 13 tasks. Overall, FT-Agent achieves the best performance on **10 out of 13 tasks**, establishing a strong initial autonomous baseline for FT-Dojo. The multi-run results show that the main qualitative conclusions hold despite stochastic variation, with larger variance on chemistry reaction similarity metrics.

**Breakthroughs on Sparse Supervision.** The most striking result is observed on the **AIME 2025** task. As detailed in Table 1, 82% of raw training samples lack solutions and provide no Chain-of-Thought (CoT) rationale. Consequently, the base model, tool-augmented OpenHands, and constrained Manual SFT baseline all achieve 0% accuracy. FT-Agent reaches **11.11%** on average across three runs, indicating that the agent can identify missing reasoning supervision as a bottleneck and synthesize useful training trajectories under the fixed resource budget.

**Proficiency in Code and Structure.** Beyond pure reasoning, FT-Agent is strongest on tasks requiring executable code generation and strict structural constraints. In Table QA Visualization, FT-Agent achieves a Pass@1 of **29.33%**, outperforming the tool-augmented OpenHands baseline (24.00%). Similarly, in Chemistry Molecule Editing, it attains **54.44%** accuracy, outperforming OpenHands

*Table 2.* **Main results comparison.** We compare FT-Agent against baselines across 13 tasks in 5 domains. FT-Agent achieves the best performance on 10 out of 13 tasks. **Bold** indicates best performance. FT-Agent results are reported as mean±std over 3 independent runs. Manual SFT is a constrained, resource-matched manual baseline without LLM assistance. See Appendix A for metric definitions and abbreviations.

| | Math | Patent Examination | | | Table QA | | | | Finance | Chemistry | | | | | | | |
| | AIME | PAR4PC | NOC4PC | PI4PC | DA | FC | NR | Vis. | QA | | Mol_Und | | Mol_Edit | Mol_Opt | | Reaction | |
| Method | Acc↑ | F1↑ | F1↑ | Acc↑ | Acc↑ | Acc↑ | Acc↑ | P@1↑ | Acc↑ | MAE↓ | TMS↑ | Acc↑ | Acc↑ | SR↑ | VS↑ | FTS↑ | Acc↑ |
|---|---|---|---|---|---|---|---|---|---|---|---|---|---|---|---|---|---|
| Qwen2.5-7B-Instruct | 0.00 | 57.78 | 26.20 | 34.00 | 32.78 | 75.00 | 32.00 | 4.00 | 65.10 | 0.53 | 0.09 | 63.67 | 22.20 | 23.33 | 68.67 | 15.59 | 30.00 |
| Manual SFT | 0.00 | **67.61** | 33.39 | 28.00 | 28.80 | 70.83 | 43.00 | 8.00 | **71.68** | 0.65 | 0.00 | 36.67 | 0.00 | 9.67 | 31.00 | 0.00 | 10.00 |
| OpenHands | 0.00 | 50.51 | 28.44 | 44.00 | 23.22 | 66.67 | 40.00 | 24.00 | 64.73 | 0.65 | 0.10 | **68.00** | 40.00 | 31.00 | 80.67 | **35.59** | **60.00** |
| FT-Agent | **11.11**±3.82 | **67.61**±2.40 | **36.81**±4.13 | **49.00**±4.36 | **34.71**±0.28 | **81.25**±2.08 | **45.33**±1.15 | **29.33**±6.11 | 67.20±0.31 | **0.23**±0.16 | **0.36**±0.06 | 67.55±5.68 | **54.44**±2.94 | **34.44**±2.17 | **90.00**±4.41 | 30.92±11.22 | 35.33±3.06 |

(40.00%). Manual SFT remains competitive in standardized formats such as Financial QA, which suggests that autonomous fine-tuning is most useful when the task requires iterative data diagnosis, dynamic tool use, or constraint-aware optimization. Table 4 includes the LLM-assisted Manual SFT variant as a five-task reference.

*Table 3.* Exploration dynamics between OpenHands and FT-Agent.

| Metric | OpenHands | FT-Agent |
|---|---|---|
| Avg. Loops | 3.69 | 8.77 |
| Avg. Improve Rate | 17.31% | 24.37% |
| Avg. Cost (iterative only) | $3.92 | $5.73 |

**Exploration Depth and Efficiency.** Table 3 shows that FT-Agent sustains deeper and more effective exploration than OpenHands, with more loops (8.77 vs 3.69) and a higher improvement rate (24.37% vs 17.31%) under the same toolset. The modest extra iterative cost ($5.73 vs $3.92) reflects a trade-off for tasks where shallow exploration stalls. These patterns suggest that FT-Agent is most valuable when data is imperfect (e.g., AIME) or constraints are rigid (e.g., Chemistry), while simpler baselines can remain competitive on cleaner standardized tasks.

### 4.3. Frontier Agents and Backbone Generality

Reviewers raised whether FT-Agent's gains reflect fine-tuning-oriented design or simply a stronger compared agent framework/backbone. We therefore add controlled comparisons on the five representative tasks used in our ablation study. All autonomous systems use the same FT-Dojo task setup, Docker environment, LlamaFactory/OpenCompass wrappers, data repository, and 12-hour wall-clock budget. We also include the LLM-assisted Manual SFT result as an assisted-human reference on the same five tasks.

FT-Agent with GPT-5.2 achieves the highest average among all autonomous systems and the assisted-human reference. Among GPT-5.2-based agents, it outperforms both Open-Hands and Codex, suggesting that the bottleneck is not only code generation quality but also data-quality diagnosis and multi-level feedback interpretation. The LLM-assisted Manual SFT row performs best on AIME but requires human

data-synthesis decisions, so we use it to scope the constrained manual comparison rather than to replace the full 13-task baseline. Open-source FT-Agent variants remain competitive, with DeepSeek-V3.2 surpassing OpenHands and approaching Codex on average. These results support FT-Agent's role as a strong fine-tuning-oriented baseline rather than establishing it as a universally superior agent framework.

**Budget Sensitivity.** To test whether the 12-hour wall-clock budget disproportionately favors fail-fast designs, we also ran OpenHands on AIME 2025 with a 24-hour budget across two runs. We observed no meaningful improvement over the 12-hour setting: in both runs, the best checkpoint appeared within the first 6 hours, and later iterations did not improve the final score. This suggests that simply extending time does not address the main failure mode when the agent does not diagnose missing CoT supervision as the bottleneck.

### 4.4. Ablation Study

We conduct a comprehensive ablation study to disentangle the impact of training data size, planning backbone, and target model scale. We select five representative tasks covering diverse modalities: reasoning (AIME), retrieval (Patent-PI4PC), structural constraints (Chem-Mol_Edit), knowledge (Financial QA), and code generation (Table-Vis). Results are summarized in Figure 3.

**1. Quality over Quantity (Data Scaling).** Expanding the training set from 2k to 5k samples yields mixed results. While performance improves on pattern-intensive tasks like Patent Retrieval and Molecule Editing, it degrades on Visualization. Under a fixed 12-hour budget, larger datasets consume significantly more compute time for processing and training, squeezing the window for strategy iteration. The 2k budget strikes a better balance between data coverage and exploration depth.

**2. Necessity of Frontier Intelligence (Backbone Sensitivity).** We substitute the planning backbone in the iterative loop while keeping the underlying data-cleaning models fixed. GPT-5.2 achieves the best average among FT-Agent variants, while GPT-4o improves Finance QA but drops on procedural tasks such as Visualization and PI4PC; open-

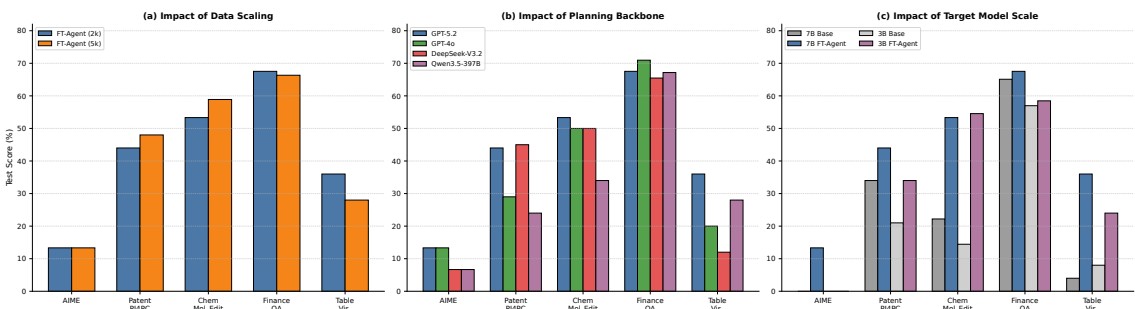

*Figure 3.* **Ablation study results.** (a) *Data scaling*: Performance comparison between 2k and 5k training samples. (b) *Planning backbone*: FT-Agent variants with proprietary/open-source planners. (c) *Target model scale*: Performance across target sizes. Panel (b) shows FT-Agent variants only; cross-agent baselines are in Table 4, with full ablations in Appendix Table 10.

*Table 4.* **Five-task controlled comparison.** Frontier agents, FT-Agent planning backbones, and an assisted-human reference on representative tasks. Scores are percentages; Avg. is over displayed tasks.

| Method | Setting | AIME | PI4PC | Mol_Edit | Finance | Vis. | Avg. |
|---|---|---|---|---|---|---|---|
| Base Model | – | 0.00 | 34.00 | 22.20 | 65.10 | 4.00 | 25.06 |
| Manual SFT | LLM synthesis | **20.00** | 28.00 | 50.00 | 67.57 | 20.00 | 37.11 |
| OpenHands | GPT-5.2 | 0.00 | 44.00 | 40.00 | 64.73 | 24.00 | 34.55 |
| Codex | GPT-5.2 | 6.67 | 42.00 | 40.00 | 65.63 | 26.00 | 36.06 |
| Claude Code | Sonnet-4.6-thinking | 0.00 | **51.00** | 44.00 | 68.32 | **36.00** | 39.86 |
| FT-Agent | Qwen3.5-397B-A17B | 6.67 | 24.00 | 34.00 | 67.16 | 28.00 | 31.97 |
| FT-Agent | DeepSeek-V3.2 | 6.67 | 45.00 | 50.00 | 65.47 | 12.00 | 35.83 |
| FT-Agent | GPT-4o | 13.33 | 29.00 | 50.00 | **70.95** | 20.00 | 36.65 |
| FT-Agent | GPT-5.2 | 13.33 | 44.00 | **53.33** | 67.53 | **36.00** | **42.83** |

source backbones remain competitive but show larger task-specific variance. These results indicate that the framework transfers across planners, but high-level orchestration quality remains critical for navigating the optimization landscape effectively.

**3. Robustness Across Tested Model Scales.** FT-Agent demonstrates strong generalizability, consistently unlocking significant gains on the smaller 3B model (e.g., +40% on Mol_Edit, +16% on Visualization). These results suggest that FT-Agent can transfer across the tested target model scales: its ability to diagnose weaknesses and synthesize data remains effective on both 3B and 7B backbones under our benchmark setting. Broader coverage over larger model families and training regimes remains an important direction for future work.

### 4.5. Case Study

To understand the cognitive mechanisms governing these outcomes, we analyze the step-by-step decision traces of FT-Agent (detailed in Appendix D). This analysis reveals a stark duality: the capacity for **cumulative learning from historical experience** versus the persistence of **causal reasoning deficits**.

**Cumulative Learning from Historical Experience.** The

trajectory on **Mol_Edit** (Figure 4a) exemplifies the agent's capacity for iterative self-improvement through historical memory. Starting from a catastrophic failure (2% due to vocabulary corruption), the agent diagnosed and addressed issues across iterations: correcting data format misalignment, introducing RDKit validation to filter invalid SMILES, fixing insufficient training steps, and aligning operation distributions with the benchmark. This progression ultimately drove accuracy to 56%, demonstrating how cumulative learning enables increasingly robust solutions.

**The Causal Reasoning Gap.** Conversely, **PI4PC** (Figure 4b) illustrates the "shotgun debugging" failure mode. Facing stagnation, the agent indiscriminately applied generic techniques like NEFTune without causal diagnosis. These ungrounded interventions caused severe volatility (collapsing to 8%) and yielded zero net improvement. This highlights a critical limitation: while current agents are efficient *tool optimizers*, they struggle to act as rigorous *scientists* when hypothesis testing requires deep causal insight.

## 5. Related Work

**Interactive Environments for Agents.** Recent benchmarks have shifted from static dataset evaluation to interactive, environment-based assessments. In software engineering,

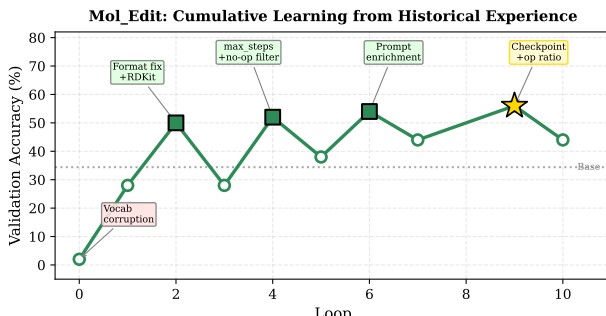
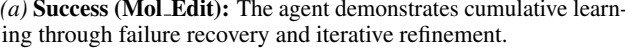

*(a)* **Success (Mol_Edit):** The agent demonstrates cumulative learning through failure recovery and iterative refinement.

*(b)* **Failure (PI4PC):** The agent struggles with causal diagnosis, yielding zero net gain.

*Figure 4.* **Contrasting Learning Trajectories.** (a) On Chemistry, the agent demonstrates *cumulative learning*, recovering from failure and progressively refining its approach through domain tool integration and iterative optimization. (b) On Patent Classification, it exhibits *shotgun debugging*, cycling through advanced techniques without identifying the root cause of overfitting.

SWE-Bench and SWE-agent demonstrated the potential of agents to resolve real-world GitHub issues (Jimenez et al., 2023; Yang et al., 2024). In the machine learning domain, MLE-bench and MLE-Dojo evaluate agents on Kaggle-style competitions (Chan et al., 2024; Qiang et al., 2025). Recent work has also begun to study how MLE agents should optimize, with Gome (Zhang et al., 2026) framing structured diagnostic reasoning as a gradient-like signal for directed updates beyond scalar-score tree search. However, these benchmarks typically operate under fixed-data assumptions, confining agents to feature engineering or model searching. FT-Dojo distinguishes itself by formalizing the end-to-end fine-tuning workflow, where agents must navigate an open-ended decision space involving raw data curation, processing logic, and training configuration.

**Automated LLM Training Configuration.** Implementing fine-tuning pipelines from scratch presents a significant barrier for automation, requiring complex boilerplate code for distributed training, model loading, and gradient synchronization. To lower this barrier, integrated frameworks have emerged (Zheng et al., 2024; Zhao et al., 2024), abstracting these complexities into unified, configuration-driven interfaces (e.g., YAML or JSON). Building on these abstractions, recent works such as LaMDAgent (Yano et al., 2025) utilize agents to optimize post-training workflows. However, these approaches typically rely on predefined templates or blind search, lacking the adaptability to dynamically tailor configurations to specific dataset characteristics (e.g., sequence length) or hardware constraints (e.g., memory limits). FT-Agent bridges this gap by autonomously interfacing with training frameworks to derive optimal configurations directly from environmental and data contexts.

**Data-Centric LLM Fine-Tuning.** Alongside training infrastructure, data quality is recognized as a pivotal factor in model adaptation. Recent tooling has focused on standard-izing processing pipelines; for instance, Data-Juicer (Chen et al., 2024a;b) and EasyDataset (Miao et al., 2025) provide composable operators for cleaning and formatting. Beyond processing, algorithmic approaches like DoReMi (Xie et al., 2023) and LESS (Xia et al., 2024) introduce metrics to efficiently select influential samples or optimize data mixtures. However, these methods operate in an *open-loop* manner using static metrics, failing to address specific capability gaps exposed during training. FT-Agent instead treats data curation as an iterative optimization problem. It autonomously formulates targeted strategies, such as synthesizing missing concepts or re-weighting hard examples, to resolve actual failure modes based on evaluation feedback.

## 6. Conclusion

In this work, we introduced **FT-Dojo**, an interactive benchmark environment for evaluating autonomous agents on end-to-end LLM fine-tuning. Across 13 tasks in 5 domains, FT-Dojo requires agents to select and construct training data from raw sources, configure training pipelines, and refine their strategies from structured feedback, making data strategy and training configuration joint optimization targets rather than fixed inputs.

We further proposed **FT-Agent**, a fine-tuning-oriented framework built around structured iteration planning, fail-fast validation, and multi-level feedback analysis. Experiments show that FT-Agent provides a strong initial baseline, achieving the best performance on 10 of 13 tasks and remaining competitive in controlled comparisons with frontier agents and open-source planning backbones. Our case studies further show both the promise and limits of autonomous fine-tuning: agents can recover from failures through historical experience, but still struggle with causal diagnosis and long-horizon planning, leaving open-world data sourcing and stronger reasoning mechanisms as future work.

## Impact Statement

This paper introduces FT-Dojo, an interactive benchmark, and FT-Agent, an autonomous agent framework for end-to-end LLM fine-tuning. By automating the fine-tuning pipeline—from data curation to model evaluation—our work has the potential to lower the barrier for adapting LLMs to specialized domains, broadening access to domain-specific AI capabilities. However, reducing the human effort required for fine-tuning also lowers the cost of producing models tailored for harmful purposes, such as generating misleading content in sensitive domains like finance or law. We believe that the transparency of our benchmark design and the emphasis on structured evaluation mitigate some of these risks by encouraging reproducible and auditable fine-tuning practices. We encourage future work to incorporate safety-oriented evaluation criteria within similar benchmarks.

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

# Appendix

# A. Data Sources and Benchmark Details of FT-Dojo

This section elaborates on the construction details of the FT-Dojo suite. We provide a comprehensive overview of the evaluation benchmarks, the corresponding training resources, our rigorous data processing pipeline, and detailed statistics for each subtask.

## A.1. Evaluation Benchmarks

### A.1.1. AIME 2025 (MATHEMATICS)

**Source:** American Invitational Mathematics Examination 2025 (Art of Problem Solving Wiki, 2025).
AIME 2025 serves as our primary testbed for high-difficulty mathematical reasoning. It contains competition-level problems requiring multi-step derivation across algebraic manipulation, number theory, combinatorics, and geometry. Unlike standard benchmarks, problems demand error-sensitive symbolic manipulation under strict exact-match evaluations, where answers are integers between 0–999. *Metrics: Accuracy*, using a cascade evaluator that first applies symbolic verification and then sends unresolved cases to a fixed LLM judge with the rubric in Appendix A.4.

### A.1.2. PANORAMA (PATENT EXAMINATION)

**Source:** LG AI Research & KAIST (Lim et al., 2025).
PANORAMA simulates the professional patent examination process. It is constructed from U.S. patent records preserving complete "decision trails" (applications, cited prior art, office actions). We evaluate on three subtasks:

- **PAR4PC (Prior Art Retrieval):** Given a claim and 8 candidate patents, identify the specific prior art cited for rejection. *Metrics: Accuracy (Exact Set Match), Macro-F1.*

- **NOC4PC (Novelty/Obviousness Classification):** Classify claims as *ALLOW*, *§102* Rejection (Lacks Novelty), or *§103* Rejection (Obvious). *Metrics: Accuracy, Macro-F1.*

- **PI4PC (Paragraph Identification):** Identify the precise paragraph within a prior art document that supports the rejection. *Metrics: Accuracy.*

### A.1.3. CHEMCOTBENCH (CHEMISTRY)

**Source:** ChemCoTBench (Li et al., 2025).
Constructed from a massive corpus of 2 million molecular samples, ChemCoTBench represents a shift from traditional knowledge retrieval to generative chemical reasoning. Unlike simple QA tasks, this benchmark evaluates an LLM's ability to perform precise, multi-step molecular manipulations under strict physical constraints. It introduces a Modular Chain-of-Thought (CoT) framework, where complex tasks (e.g., optimizing drug solubility while maintaining scaffold integrity) are decomposed into discrete, verifiable operations. Crucially, the benchmark enables intermediate consistency checking, allowing automatic verification of chemical validity (e.g., valence rules, ring integrity) at each reasoning step, rather than relying solely on the final answer. The suite covers the full lifecycle of chemical discovery:

- **Mol_Und (Molecular Understanding):** Tasks include functional group counting, ring counting, and SMILES equivalence. *Metrics: MAE (for counting), Tanimoto Similarity, Accuracy.*

- **Mol_Edit (Molecular Editing):** Structural modification (add/delete/substitute) while maintaining validity. *Metrics: Accuracy, Tanimoto Similarity.*

- **Mol_Opt (Molecular Optimization):** optimizing molecules for properties (e.g., QED, LogP, DRD2) while preserving scaffolds. *Metrics: Success Rate, Valid SMILES %, Scaffold Match.*

- **Reaction:** Forward/retrosynthesis and condition recommendation. *Metrics: Fingerprint Tanimoto Similarity, BLEU, Accuracy.*

### A.1.4. FINANCEIQ (FINANCE)

**Source:** Chinese financial certification examination dataset (Duxiaoman-DI, 2023).
Covers 10 Chinese financial certification exams (CPA, banking, securities, fund, insurance, economic analyst, taxation, futures, financial planner, actuarial). Tests precision under numeric pressure and multi-step financial reasoning with finance-specific assumptions. Task: Multiple-choice questions. *Metrics: Accuracy*.

### A.1.5. TABLEBENCH (TABLE QA)

**Source:** TableBench (Wu et al., 2025).
A comprehensive benchmark for structure-aware grounding over tabular evidence, consisting of 886 test cases across 18 fields.

- **DA (Data Analysis):** Trend forecasting and statistical analysis ($\pm$10% tolerance). *Metrics: Accuracy.*

- **FC (Fact Checking):** Verifying statements against tabular data. *Metrics: Exact Match.*

- **NR (Numerical Reasoning):** Arithmetic operations (sum, average, percentage). *Metrics: Accuracy.*

- **Vis (Visualization):** Generating Python code to plot charts. *Metrics: Pass@1 (Executable Code Rate).*

### A.2. FT-Dojo Raw Training Resources

To facilitate reproducible research and model adaptation, FT-Dojo incorporates a curated repository of **raw training data** spanning all five target domains. We aggregated these resources from diverse high-quality public sources to serve as the foundational curriculum (or seed supervision) for agent fine-tuning. Prior to inclusion, all raw data underwent a rigorous decontamination process to strictly eliminate any overlap with the FT-Dojo evaluation benchmarks, ensuring the integrity of our zero-shot assessment.

### A.2.1. MATHEMATICS: DEEPSCALER

**Source:** agentica-org/DeepScaleR-Preview-Dataset.
DeepScaleR contains approximately 40,000 competition-level mathematics problems compiled from AIME (1984–2023), AMC, Omni-MATH, and Still datasets. Problems cover algebraic manipulation, number theory, combinatorics, and geometric reasoning.

**Data Quality:** 82% of samples lack solutions; remaining 18% contain brief summary-style answers. CoT generation is required for effective training.

### A.2.2. PATENT EXAMINATION: PANORAMA

**Source:** LG-AI-Research/PANORAMA (Lim et al., 2025).
We use the official training split containing 254,449 samples across three tasks: Prior Art Retrieval (54,028), Novelty Classification (136,211), and Paragraph Identification (64,210).

**Data Quality:** Q&A pairs only, no chain-of-thought annotations. We remove validation and test splits to prevent data leakage.

### A.2.3. CHEMISTRY: CHEMCOTDATASET

**Source:** OpenMol/ChemCoTDataset (Li et al., 2025).
The dataset provides approximately 23,000 samples with structured chain-of-thought annotations, distilled from Gemini-2.5-pro, DeepSeek-R1, and Claude-3.7-sonnet, validated by 13 chemistry PhD candidates with ¿90% accuracy.

**Data Quality:** Medium-high quality CoT annotations covering molecular understanding, editing, optimization, and reaction prediction.

A.2.4. FINANCE: FINANCEIQ

**Source:** Duxiaoman-DI/FinanceIQ (Duxiaoman-DI, 2023).
Contains 6,179 multiple-choice questions from 10 Chinese financial professional certification exams, including CPA, banking, securities, insurance, and actuarial domains.

**Data Quality:** Q&A pairs only, without reasoning chains. CoT generation recommended for optimal performance.

A.2.5. TABLE QA: TABLEINSTRUCT

**Source:** Multilingual-Multimodal-NLP/TableInstruct (Wu et al., 2025).
Large-scale instruction tuning dataset for table question answering, containing diverse examples across fact checking, numerical reasoning, data analysis, and visualization tasks.

**Data Quality:** Includes reasoning chains with multiple instruction types: Direct Prompting (DP), Textual CoT (TCoT), Program-of-Thought (PoT), and Symbolic CoT (SCoT).

## A.3. Data Processing

A.3.1. EVALUATION DATA CONSTRUCTION

To construct robust evaluation splits, we first apply context-length filtering to ensure compatibility across model architectures. We then randomly partition the filtered instances into non-overlapping validation and test sets. Specifically, for each benchmark (or subtask), we allocate samples according to:

$$N_{val} = N_{test} = \min(100, \lfloor n/2 \rfloor) \tag{3}$$

where $n$ is the total number of filtered instances. The validation set is the only held-out feedback visible during optimization; the test set remains hidden until final assessment. This strict separation ensures that agents are evaluated on held-out data distinct from their training resources.

A.3.2. TRAINING DATA PREPARATION

All training datasets are processed with the following protocol:

1. **Leakage Prevention:** For datasets with official splits (e.g., PANORAMA), validation and test partitions are explicitly removed.

2. **Format Normalization:** All instances are converted to a unified instruction-response format.

3. **Quality Filtering:** Samples exceeding context windows or containing corrupted text are filtered out.

## A.4. Evaluator Rubrics and LLM-as-a-Judge Prompt

Most FT-Dojo tasks use deterministic or task-specific evaluators, including exact match, accuracy, Macro-F1, executable-code checks, molecular validity, Tanimoto similarity, BLEU, and domain-specific chemistry metrics. The main LLM-as-a-judge case is AIME 2025. For AIME, we use a cascade evaluator: a rule-based mathematical verifier is applied first, and only unresolved cases are sent to a fixed GPT-5 judge. The judge prompt, rubric, and model are identical across all methods.

---

**AIME Judge Prompt Template**

You are grading a math answer for correctness.

**Input**: original problem, gold answer, and model prediction.

**Rubric**: Mark the prediction as correct if it is mathematically equivalent to the gold answer. Ignore formatting differences such as boxed notation, whitespace, punctuation, or explanatory text. For multi-part answers, all required parts must be correct. Do not give partial credit. If the answer is ambiguous, missing, or not mathematically equivalent, mark it incorrect.

---

> **Output**: Return exactly one label: `CORRECT` or `INCORRECT`.

## A.5. Dataset Statistics

We provide a granular breakdown of the evaluation datasets. Table 5 summarizes the sample counts for each specific subtask within ChemCoTBench, TableBench, and PANORAMA, ensuring balanced coverage of all task types (e.g., distinguishing between *Forward Synthesis* and *Retrosynthesis*).

In contrast, AIME 2025 (Mathematics) and FinanceIQ (Finance) are evaluated as holistic benchmarks. Due to their unified nature, they are not stratified into subtasks but are maintained as monolithic validation and test sets to evaluate overall domain proficiency. For AIME 2025, only 30 filtered instances remain after compatibility filtering, yielding 15 validation and 15 test instances under the split rule above. FinanceIQ follows the 100/100 validation/test allocation.

# B. Implementation Details of Baselines

To ensure a rigorous evaluation, we benchmark FT-Agent against primary baselines that represent manual fine-tuning, general-purpose autonomous coding agents, and stronger frontier-agent systems. This section details the specific implementation protocols for each.

## B.1. Manual SFT (Constrained Manual Baseline)

The Manual SFT baseline is a constrained, resource-matched manual workflow for model adaptation. It was executed by **senior NLP researchers, each possessing over five years of experience** in large language model training and data engineering. We do not treat this baseline as a human-expert upper bound or as a fully assisted human best-effort result; it is a documented comparison point under the same wall-clock budget and a restricted iteration protocol.

### B.1.1. WORKFLOW PROTOCOL

Practitioners followed a standard three-stage pipeline, mirroring the agent's logical steps but executed manually:

- **Data Engineering:** Developing custom Python pipelines (using Pandas/Datasets) to implement **robust rule-based cleaning strategies** (e.g., removing noise, deduplication) prior to converting and tokenizing the data into the standard Alpaca schema.

- **Training Configuration:** Manually configuring `train_config.yaml` for LLaMA-Factory, selecting hyperparameters (e.g., Learning Rate 1e-4 $\sim$ 5e-5, cosine scheduler) based on best practices for 7B models.

- **Evaluation:** Executing OpenCompass scripts to validate performance and iteratively refining the approach based on results.

### B.1.2. CONSTRAINTS

To match the agent's compute budget, human practitioners were restricted to a **"Two-Pass" protocol**: one initial exploration run and one refinement run. They were prohibited from performing open-ended grid searches and did not use LLM assistance for data synthesis or cleaning in the main Manual SFT baseline. This restricted iteration count is an important limitation of the comparison.

### B.1.3. LLM-ASSISTED MANUAL SFT CHECK

To assess the effect of LLM access for manual data synthesis and cleaning, we additionally ran an LLM-assisted Manual SFT variant on five representative tasks. This variant improves substantially on tasks where missing CoT or difficult data cleaning limits the original constrained protocol. FT-Agent remains competitive while operating fully autonomously, but we use these results to scope the Manual SFT comparison rather than to claim superiority over unconstrained human experts. The FT-Agent row in this table uses the representative single-run scores from the corresponding comparison; the main table reports 3-run mean and standard deviation.

*Table 5.* Detailed statistics of evaluation datasets across FT-Dojo benchmarks.

| Task Category | Subtask | Total | Valid | Test |
|---|---|---|---|---|
| **1. ChemCoTBench (Chemistry)** | | | | |
| Mol_Und | Murcko_scaffold | 40 | 20 | 20 |
| | equivalence | 100 | 50 | 50 |
| | fg_count | 100 | 50 | 50 |
| | ring_count | 20 | 10 | 10 |
| | ring_system_scaffold | 60 | 30 | 30 |
| | **Subtotal** | **320** | **160** | **160** |
| Mol_Edit | add / delete | 40 | 20 | 20 |
| | substitute | 60 | 30 | 30 |
| | **Subtotal** | **100** | **50** | **50** |
| Mol_Opt | drd / gsk / jnk | 300 | 150 | 150 |
| | logp / qed / solubility | 300 | 150 | 150 |
| | **Subtotal** | **600** | **300** | **300** |
| Reaction | Forward / Retrosynthesis | 200 | 100 | 100 |
| | Reaction Condition | 90 | 45 | 45 |
| | Next Step Product | 85 | 42 | 42 |
| | Mechanism Selection | 100 | 50 | 50 |
| | **Subtotal** | **475** | **237** | **237** |
| **2. TableBench (Table QA)** | | | | |
| Data Analysis | Anomaly / Causal / Correlation | 142 | 70 | 70 |
| | Descriptive / Impact / Stat | 151 | 75 | 75 |
| | TrendForecasting | 50 | 25 | 25 |
| | **Subtotal** | **343** | **170** | **170** |
| Fact Checking | MatchBased / Multi-hop | 96 | 48 | 48 |
| Num. Reasoning | Aggregation / Arith / Comp | 150 | 75 | 75 |
| | Count / Domain / Multi-hop | 150 | 74 | 74 |
| | Ranking / Time-based | 97 | 48 | 48 |
| | **Subtotal** | **397** | **197** | **197** |
| Visualization | ChartGeneration | 50 | 25 | 25 |
| **3. PANORAMA (Patent)** | | | | |
| Patent Exam. | Prior Art Retrieval (PAR4PC) | 2896 | 100 | 100 |
| | Novelty Classif. (NOC4PC) | 2884 | 100 | 100 |
| | Paragraph ID (PI4PC) | 3402 | 100 | 100 |
| | **Total** | **9182** | **300** | **300** |

*Table 6.* Manual SFT with LLM-assisted synthesis on five representative tasks. All scores are percentages and higher is better. FT-Agent values are representative single-run scores for this focused comparison.

| Method | AIME | PI4PC | Table Vis. | Finance QA | Mol_Edit |
|---|---|---|---|---|---|
| Manual SFT (original) | 0.00 | 28.00 | 8.00 | **71.68** | 0.00 |
| Manual SFT (w/ LLM synthesis) | **20.00** | 28.00 | 20.00 | 67.57 | 50.00 |
| FT-Agent | 13.33 | **44.00** | **36.00** | 67.53 | **53.33** |

## B.2. OpenHands Baseline (Tool-Augmented Harness)

We evaluated **OpenHands (v0.14)** using the **CodeAct** architecture. To ensure a fair comparison, we implemented a specialized evaluation harness that standardizes the execution environment.

### B.2.1. EXPERIMENTAL SETUP

We evaluate OpenHands on the FT-Dojo benchmark under conditions strictly identical to FT-Agent. To isolate the impact of the framework design, we enforce a rigorous control protocol where both agents operate on a single **NVIDIA B200 GPU** with a strict **12-hour wall-clock limit** and use the same base model (**Qwen2.5-7B-Instruct**) for fine-tuning.

Furthermore, we align all external resources:

- **Reasoning Engine:** Both agents invoke identical LLM API endpoints (**GPT-5.2**) for reasoning.

- **Data Access:** Both access the exact same Data Repository $\mathcal{D}$ (raw training files) without prior processing.

- **Environment:** Both operate within the same Dockerized sandbox with standard Python/Bash utilities.

OpenHands is prompted with the same high-level task descriptions and has access to the same fine-tuning/evaluation wrappers, but it operates without FT-Agent's structured iteration planning, fail-fast validation, best-configuration tracking, and structured feedback analysis.

### B.2.2. TOOL-AUGMENTED ENVIRONMENT

To prevent the baseline from failing due to trivial CLI syntax errors, we injected two high-level custom tools into the sandbox:

- **LlamaFactoryTool:** A Python wrapper around `llamafactory-cli`. It accepts structured parameters (e.g., `finetuning_type='lora'`) via a Pydantic interface, automatically generating valid `train_config.yaml` and parsing training logs.

- **OpenCompassTool:** A wrapper for the evaluation stack using Jinja2 templates to dynamically generate configurations and parse CSV summaries into structured JSON observations.

### B.2.3. SCAFFOLDED WORKFLOW ORCHESTRATION

Unlike FT-Agent which autonomously plans its research trajectory, we design a systematic scaffold in `main.py` to guide the OpenHands baseline through standard research stages:

1. **Data Cleaning Stage:** The harness instantiates a `DataAgent` with a fixed prompt requesting a cleaning script. It includes a "Self-Correction Loop" that feeds `stderr` back to the agent upon failure.

2. **Training Stage:** The harness transitions to a `TrainAgent` equipped with `LlamaFactoryTool`, providing heuristic parameter guidelines (e.g., based on sample size).

3. **Evaluation Stage:** The harness invokes an `EvalAgent` to run `OpenCompassTool`.

### B.2.4. Resilience Mechanisms

To prevent failure from transient network issues, the harness includes enterprise-grade resilience features:

- **Load Balancing:** API requests are distributed across multiple local endpoints (ports 4000–4007).

- **Transient Retry:** A `run_conversation_with_retry` wrapper automatically handles LLM API timeouts with linear backoff.

### B.3. Frontier-Agent and Open-Source Backbone Baselines

For the additional comparison in Table 4, we evaluate Codex (GPT-5.2) and Claude Code (Claude Sonnet-4.6-thinking) as frontier-agent baselines. We also evaluate FT-Agent with two open-source planning backbones, DeepSeek-V3.2 and Qwen3.5-397B-A17B. All systems are run on the same five representative tasks used in the ablation study: AIME, PI4PC, Mol_Edit, Finance QA, and Table Visualization.

All runs use identical external conditions: the same FT-Dojo data repository, task specifications, Docker sandbox, LlamaFactory/OpenCompass wrappers, target base model, validation/test splits, and 12-hour wall-clock budget. Frontier coding agents are given the same high-level task objective and access to the same training/evaluation commands available to OpenHands. FT-Agent variants differ only in the planning backbone; the fail-fast validation, feedback aggregation, and fine-tuning wrappers are held fixed.

### B.4. Extended-Budget Check

To examine whether the fixed 12-hour budget inherently favors fail-fast validation, we doubled OpenHands' budget to 24 hours on AIME 2025 for two independent runs. The additional time did not yield a meaningful improvement: the best checkpoint in both runs appeared within the first 6 hours, and later iterations did not improve the final score. This supports the interpretation that OpenHands' main bottleneck on AIME is not raw wall-clock time but failure to diagnose the missing-CoT supervision issue in the raw training repository.

## C. Detailed Comparison: OpenHands vs. FT-Agent

This section provides a granular analysis of why general-purpose coding agents, exemplified by OpenHands (Wang et al., 2024), struggle with the specialized domain of autonomous LLM fine-tuning. Crucially, this performance gap persists even when OpenHands is equipped with the identical toolset (LlamaFactory/OpenCompass wrappers) as FT-Agent, isolating the root cause to cognitive architecture rather than tool availability.

### C.1. Quantitative Results Comparison

To substantiate the cognitive disparity between the two agents, Table 7 presents the comprehensive evaluation results across all 13 tasks. Unlike the condensed view in the main text, this table reports performance on both **Validation** and **Test** splits, alongside auxiliary metrics (e.g., Macro F1, Tanimoto Similarity) to provide a holistic assessment of agent capability.

**Robustness and Generalization**   Crucially, the performance gains of FT-Agent are consistent across validation and test sets. For instance, in the **Patent Examination** domain, FT-Agent maintains a clear lead in both Exact Match and F1 scores across splits, mitigating potential concerns of test-set overfitting. Similarly, in the **Math** domain (AIME), FT-Agent achieves a non-trivial success rate (13.30% Test) where the baseline fails completely (0.00%), indicating that the performance gap stems from a fundamental difference in problem-solving strategy rather than stochastic variance.

**Metric-Level Granularity**   The inclusion of granular metrics reveals specific behavioral differences:

- **Structure vs. Sequence:** In **Chemistry**, FT-Agent outperforms the baseline in tasks requiring structural understanding (Mol_Und and Mol_Opt), evidenced by lower error rates (MAE) and higher validity scores. This aligns with the hypothesis that our agent effectively leverages tools to verify chemical validity.

- **Trade-offs:** Conversely, OpenHands retains a slight edge in the **Reaction** prediction task (higher BLEU). This suggests

*Table 7.* **Full performance comparison on Validation and Test sets.** This table supplements the main text by including all auxiliary metrics (e.g., Macro F1, Tanimoto, BLEU) and validation splits to demonstrate robustness. **Bold** indicates the best performance in each column (for error metrics like MAE, lower is better).

| Domain | Task | Metric | OpenHands | | FT-Agent | |
|---|---|---|---|---|---|---|
| | | | **Val** | **Test** | **Val** | **Test** |
| Math | AIME | Acc | 0.00 | 0.00 | **20.00** | **13.30** |
| Patent Exam. | PAR4PC | Exact Match | 33.00 | 34.00 | **54.00** | **50.00** |
| | | Macro F1 | 50.93 | 50.51 | **75.43** | **70.34** |
| | NOC4PC | Acc | 35.35 | 34.34 | **40.00** | **44.00** |
| | | Macro F1 | 27.47 | 28.44 | **37.18** | **41.09** |
| | PI4PC | Acc | **47.00** | **44.00** | 40.00 | 44.00 |
| Table QA | DA | Acc | **33.78** | 23.22 | 33.66 | **34.48** |
| | FC | Acc | 79.17 | 66.67 | **85.42** | **83.33** |
| | NR | Acc | 45.00 | 40.00 | **52.00** | **46.00** |
| | Vis | Pass@1 | 8.00 | 24.00 | **20.00** | **36.00** |
| | | ECR@1 | **56.00** | **60.00** | 48.00 | **60.00** |
| Finance | QA | Acc | 74.40 | 64.73 | **75.00** | **67.53** |
| Chemistry | Mol_Und | MAE FG↓ | **0.18** | 0.10 | 0.26 | **0.04** |
| | | MAE Ring↓ | 0.90 | 1.20 | **0.60** | **0.80** |
| | | Tanimoto | 0.08 | 0.10 | **0.35** | **0.42** |
| | | Acc Equiv | 48.00 | **56.00** | 56.00 | 50.00 |
| | | Acc Scaffold | 70.00 | **80.00** | 83.33 | 73.33 |
| | Mol_Edit | Acc | 55.56 | 40.00 | **62.22** | **53.33** |
| | | Tanimoto | **52.63** | 53.29 | 50.11 | **57.19** |
| | Mol_Opt | SR | 34.67 | 31.00 | **38.33** | **36.67** |
| | | Valid Rate | 81.67 | 80.67 | **89.67** | **91.67** |
| | | Scaffold | 25.67 | 20.67 | **32.00** | **33.00** |
| | Reaction | FTS | **29.97** | **35.59** | 16.89 | 18.11 |
| | | BLEU | 33.16 | **44.93** | **33.79** | 33.10 |
| | | Acc | **64.00** | **60.00** | 36.00 | 38.00 |

that while FT-Agent's reasoning architecture excels at structured, rule-bound scientific tasks, the baseline's direct approach remains competitive for pure sequence-to-sequence text generation tasks.

## C.2. Efficiency Analysis: The Cost of Trial-and-Error

Efficiency in autonomous LLM fine-tuning is not merely about inference speed, but about the **success rate of experimental trials**. Table 8 compares the number of effective iterations (defined as training runs that complete the full pipeline without fatal runtime errors) within a fixed 12-hour budget.

FT-Agent achieves approximately **2.2× more effective iterations** (6.0 vs. 2.75) than OpenHands. This disparity stems from the high "opportunity cost" of failure in OpenHands:

- **Fail-Slow vs. Fail-Fast:** OpenHands often generates syntactically correct but semantically flawed training scripts (e.g., tensor shape mismatches or OOM errors) that crash only after hours of execution. FT-Agent utilizes *Progressive Validation* to verify data integrity and configuration validity *before* allocating GPU resources, ensuring that most execution time is spent on valid training runs.

- **Exploration Efficiency:** In complex tasks like *Table QA FC*, where data formatting is brittle, OpenHands wasted significant time debugging basic file parsing errors (yielding only 2 effective runs). In contrast, FT-Agent's specialized validation and orchestration pipeline resolved these formatting issues pre-training, allowing for 7 complete experimental loops.

*Table 8.* Number of effective iterations (completing full training) within the 12-hour budget. FT-Agent achieves 2.2× more effective iterations due to fail-fast validation.

| Task | OpenHands | FT-Agent |
|---|---|---|
| AIME | 2 | 5 |
| PAR4PC | 3 | 6 |
| Table QA FC | 2 | 7 |
| Mol_Opt | 4 | 6 |
| **Average** | **2.75** | **6.0** |

## C.3. Qualitative Analysis: The Cognitive Gap

The quantitative gap is a symptom of a fundamental difference in cognitive architecture. While both agents utilize the same underlying toolset (LlamaFactory), their approach to the problem space differs radically:

### C.3.1. OPENHANDS: THE GENERALIST TRAP

OpenHands treats fine-tuning as a generic software engineering task.

- **Blind Code Generation:** It generates data processing scripts and training configurations as arbitrary code. Lacking domain knowledge, it frequently hallucinates non-existent hyperparameters or uses deprecated library arguments, leading to "silent failures" where training runs but yields no learning.

- **Reactive Debugging:** When errors occur, OpenHands attempts to fix the Python syntax rather than the underlying logic (e.g., fixing a JSON error instead of adjusting the learning rate), often entering infinite debugging loops.

### C.3.2. FT-AGENT: DOMAIN-SPECIFIC REASONING

FT-Agent is architected specifically for the lifecycle of model adaptation, addressing the unique challenges of autonomous LLM fine-tuning:

- **Schema-Aware Configuration:** Instead of guessing parameters, FT-Agent grounds its generation in the strict schema of the training framework, substantially reducing configuration errors.

- **Task-Aware Data Processing:** It proactively analyzes the raw data distribution (e.g., sequence length, label balance) to tailor the preprocessing strategy, preventing data-induced crashes.

- **Cumulative Learning:** Unlike the stateless retries of OpenHands, FT-Agent maintains a history of failed experiments. If a specific learning rate leads to divergence, it updates its internal belief state to avoid that region in subsequent iterations.

This analysis validates the design principles introduced in §3.2: effective autonomous LLM fine-tuning requires specialized mechanisms, specifically verification, structural grounding, and historical reflection, that extend beyond the capabilities of general-purpose code generation.

## D. Case Study: Emergent Capabilities and Limitations

Beyond the focused optimization of individual benchmarks, we observe distinctive behavioral patterns that highlight both the **emergent intelligence** and the **cognitive boundaries** of FT-Agent. This section analyzes its capability for autonomous cross-domain transfer, its vulnerability to myopic planning, and provides a complete optimization trace on TableBench.

### D.1. Emergent Capability: Autonomous Cross-Domain Transfer

A distinctive capability of FT-Agent is its ability to autonomously identify and leverage data sources beyond the target task's primary dataset—a strategy that addresses the open-ended data decision space. We observe this behavior in two complementary forms.

*Figure 5.* Cross-domain data strategies autonomously discovered by FT-Agent. Left: for Numerical Reasoning, the agent mixes TableInstruct (90%) with out-of-domain DeepScaleR (10%). Right: for Mol_Opt, the agent combines three chemistry sub-tasks with task-informed proportions.

On TableBench Numerical Reasoning (Figure 5, left), the agent diagnosed that the core bottleneck was *arithmetic reliability* rather than table comprehension. In Loop 1, it autonomously mixed TableInstruct (1,800 samples, 90%) with DeepScaleR (200 samples, 10%)—a competition-level mathematics dataset (AIME, AMC) entirely outside the table QA domain—and applied a unified instruction prompt to bridge the format gap. This cross-domain augmentation yielded 46.00% accuracy, a 14-point gain over the 32.00% baseline.

On ChemCOT Mol_Opt (Figure 5, right), the agent employed a *cross-sub-task* variant of the same principle. By Loop 4, it recognized that molecular optimization requires understanding molecular structure (from Mol_Und) and editing operations (from Mol_Edit), constructing a composite training set of mol_opt (1,400 samples, 70%), mol_edit (500 samples, 25%), and mol_und (100 samples, 5%). This yielded a success rate of 36.67% with 91.67% valid SMILES, compared to 23.33% SR and 68.67% VS for the baseline.

Notably, this behavior is selective: for Reaction Prediction, Mol_Edit, and Mol_Und, the agent used only target-task data, indicating that FT-Agent applies domain transfer only when it identifies a specific capability gap that external data can address.

### D.2. Visualization: Data Collapse from Myopic Filtering

On the Visualization task (Figure 6), Loop 8 generated a filtering pipeline where each step passed local sanity checks: compile validation (reasonable), plot-call detection (reasonable), token limits (reasonable). Yet the agent failed to simulate their *joint* effect, which reduced the dataset from ∼1,800 to just 27 samples, a 98.5% reduction causing Pass@1 to plummet from 20% to 4%. This reveals that current LLMs optimize decisions locally without maintaining a coherent model of cumulative state changes, a form of "planning myopia" that distinguishes them from deliberative reasoners capable of multi-step lookahead.

### D.3. Evidence for Pattern-Matching: PI4PC Decision Trace

To substantiate our claim that agents exhibit pattern-matching rather than causal reasoning, we present the complete decision trajectory for PI4PC, including the agent's hypotheses and the resulting performance changes.

**Analysis.** Table 9 reveals a consistent pattern: the agent *describes* what went wrong but does not *explain* why at a mechanistic level. Several observations:

- **Hedging language**: The agent uses "likely," "appears," "can" rather than definitive causal claims, indicating it is *guessing* at correlations rather than *verifying* mechanisms.
- **Symptom–remedy pattern**: Each loop follows "observe symptom → apply generic ML technique" without asking *why* the symptom occurred for this specific task.
- **Insights not operationalized**: Loop 2 correctly identifies "train–eval objective mismatch," yet subsequent loops (3–7)

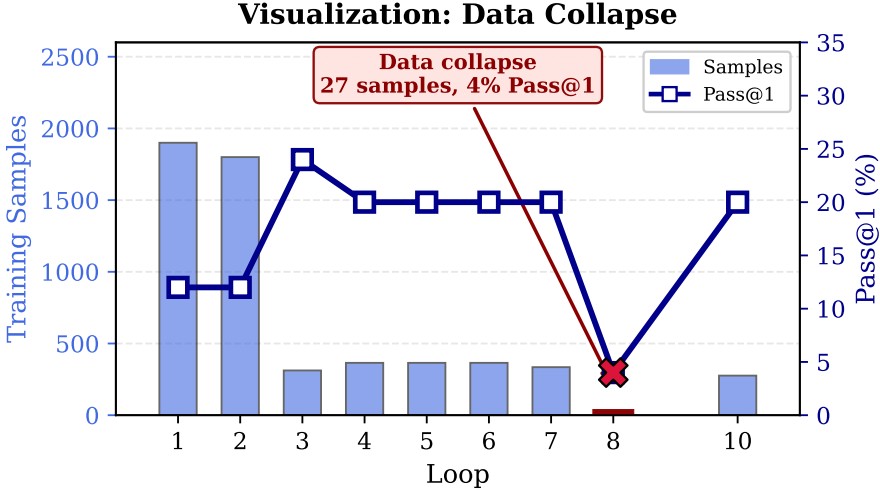

*Figure 6.* Visualization: training samples and Pass@1 over loops. Loop 8's multi-stage filtering pipeline reduced training data from ∼1,800 to just 27 samples, causing Pass@1 to collapse from 20% to 4%.

never design a targeted fix for this specific issue.
- **Missing causal questions**: The agent never asks: Why does PI4PC overfit with 2k samples? What links rationale fluency to paragraph selection? How does lexical overlap proxy actual task difficulty?

**Contrast with Mol_Edit.** On Mol_Edit, Loop 4's introduction of RDKit validation succeeded precisely because it addressed the *causal* bottleneck: invalid SMILES strings were causing silent evaluation failures. This domain-specific intervention emerged not from explicit causal reasoning but from the agent's exposure to chemistry-related training data that associated molecular tasks with RDKit. The distinction illustrates that agents can occasionally discover effective interventions through pattern-matching when the solution happens to be well-represented in their knowledge base, but struggle when task-specific causal analysis is required.

### D.4. TableBench: A Complete Optimization Trace

We present a detailed trace of FT-Agent optimizing the **TableBench** scenario. This subsection is structured chronologically to demonstrate the agent's progressive scientific discovery: starting from data selection, moving to context processing, determining training architecture, and finally refining data composition to address orthogonal capability gaps.

#### D.4.1. PROBLEM SETUP AND TASK ABSTRACTION

TableBench evaluates Large Language Models on structure-aware reasoning, demanding precise operations under strict output contracts. Upon initialization, the agent diagnoses a core mismatch: the seed dataset (`TableInstruct`) is code-heavy (Program-of-Thought), whereas the benchmark requires natural language reasoning.

#### D.4.2. INSIGHT I: STRUCTURAL COMPLEXITY DRIVES EFFECTIVE SAMPLING

*Having identified the domain mismatch, the agent's first step was to select the right training samples.*
**Observation.** Initial runs using conventional semantic stratification (e.g., *qtype*) failed to improve performance. Failure analysis revealed that errors were driven by *execution failures* (e.g., wrong row filtering) which correlated with the structural complexity of the latent program, not the question topic.

**Agent Strategy.** The agent pivoted to a **difficulty-driven sampling strategy**.

*Table 9.* PI4PC: Complete decision trajectory over 8 loops. Each row shows the agent's strategy (before execution) and feedback (after execution). Despite diverse interventions, the agent returns to baseline performance. Note the hedging language in feedback ("likely," "appears") indicating correlation-based rather than causal reasoning.

| Loop | Acc. | Strategy (Before) | Feedback (After) |
|---|---|---|---|
| 0 | 34% | Gold-anchored rationale + LoRA | "accuracy improves from 29% baseline to 34%...eval loss improving early then degrading suggests over-fitting" |
| 1 | 23% | +NEFTune + longer examiner rationales ("to reduce overfitting") | "NEFTune noise **likely** pushed the model to learn fluent justifications rather than improve discriminative paragraph choice" |
| 2 | 15% | Remove NEFTune, keep contrastive rationales | "train–eval objective mismatch where model learns plausible narratives without reliably performing discriminative selection" |
| 3 | 8% | Strict quote-matching filter ("improve data quality") | "only 1111/2000 samples survived...set **appears** extremely easy...which **can** train toward overlap shortcuts" |
| 4 | 30% | Hard-case sampling by lexical overlap | "shortfall consistent with supervision/distribution mismatch...sampled difficulty still concentrates in very-easy cases" |
| 5 | 32% | Quota-based difficulty sampler | "hard-case quota did not materialize (bucket '$\geq 7$' ended up 0)...shift **likely** didn't target hardest regimes" |
| 6 | 16% | Selector-oriented evaluation ("align train-eval") | "experiment did not actually execute the central hypothesis...used eval_loss instead of selector metric" |
| 7 | 34% | Revert to baseline configuration | "accuracy exactly matching prior SOTA (no improvement). Key hypothesis factor was not implemented" |

---

**Agent Sampling Trace**

**Insight:** Semantic labels hide execution difficulty. **Action:** Sample based on computational complexity proxies: 1. **Table Size:** (Rows × Cols) to approximate context load. 2. **Constraint Density:** Count of conjunctions/comparatives. **Goal:** Target boundary cases that challenge retrieval while remaining solvable.

---

### D.4.3. INSIGHT II: PREPROCESSING MUST BE TASK-ADAPTIVE

*With the samples selected, the agent next addressed how to fit them into the model's context window without losing information.*

**Observation.** The agent realized that context management is not a universal preprocessing step but a task-dependent variable. For **Fact Checking**, latent programs are short; for **Numerical Reasoning**, they involve long multi-step arithmetic chains.

**Agent Strategy.** The agent differentiated its pipeline based on the subtask structure:

- **Fact Checking → Truncation:** Remove redundant boilerplate to maximize batch size, as table structure is preserved.

- **Numerical Reasoning → Filtering:** Truncation is destructive to long CoT traces. Samples exceeding the context window must be filtered out to ensure reasoning consistency.

### D.4.4. INSIGHT III: RIGID FORMATTING DEMANDS FULL PARAMETER UPDATES

*Once data preparation was established, the agent had to determine the optimal learning dynamics for the training phase.*

**Observation.** While PEFT (LoRA) is standard for small datasets, TableBench requires rigid adherence to complex output formats (JSON/Exact Match).

**Agent Strategy.** The agent decided on **Full Fine-Tuning**, prioritizing "muscle memory" over parameter efficiency.

---

**Architectural Decision Trace**

**Insight:** Low-rank adapters struggle with rigid output contracts. **Rationale:** 1. **Resource:** The B200 GPU (178GB) allows full parameter updates. 2. **Performance:** Full tuning empirically yields better instruction adherence and formatting compliance. 3. **Safety:** Overfitting is mitigated via conservative learning rates rather than freezing parameters.

---

D.4.5. INSIGHT IV: ORTHOGONAL CAPABILITIES REQUIRE DATA MIXING

*Finally, in a recursive optimization loop, the agent recognized that single-source data could not cover all required capabilities.*

**Observation. Numerical Reasoning** poses a dual challenge: it requires both table grounding (finding the right numbers) and arithmetic reliability (calculating correctly). The primary source (`TableInstruct`) provided strong grounding but weak arithmetic reasoning.

**Agent Strategy.** To bridge this gap, the agent formulated a **Mixed Dataset Strategy**:

---

**Data Mixing Logic**

**Insight:** Capabilities are orthogonal. `TableInstruct` gives grounding; `DeepScaleR` gives calculation. **Action:** - **Primary (80%):** `TableInstruct` for table operations coverage. - **Auxiliary (20%):** `DeepScaleR` to inject arithmetic reliability. **Result:** A composite dataset that closes the capability gap without domain drift.

---

# E. Full Ablation Results

Table 10 presents the complete numerical results of our ablation studies. It details the performance comparisons across three key dimensions: training data scale (2k vs. 5k), agent reasoning backbone (GPT-5.2 vs. GPT-4o), and target model capacity (7B vs. 3B), covering both validation and test splits for all five benchmarks.

*Table 10.* Ablation study and baseline comparison organized by model size (7B vs. 3B). Scores are reported as **validation/test**. The **Default** setting represents FT-Agent with Qwen2.5-7B (2k samples, GPT-5.2 backbone). **Scaling-5k** and **w/ GPT-4o** are ablations on top of the 7B Default. **Bold** indicates the best validation and best test performance per row independently.

| Domain | Task (Metric) | 7B Model Settings | | | | 3B Model Settings | |
|---|---|---|---|---|---|---|---|
| | | Base | Default (2k) | Scaling-5k | w/ GPT-4o | Base | FT |
| Math | AIME25 (Acc) | 0.00/0.00 | **20.00/13.33** | 6.67/**13.33** | 6.67/**13.33** | 0.00/0.00 | 6.67/0.00 |
| Patent | PI4PC (Acc) | 29.00/34.00 | **40.00**/44.00 | 36.00/**48.00** | 38.00/29.00 | 19.00/21.00 | 31.00/34.00 |
| Finance | QA (Acc) | 73.00/65.10 | 75.00/67.53 | **75.55**/66.33 | 70.33/**70.95** | 61.44/56.98 | 60.73/58.47 |
| Chem | Mol_Edit (Acc) | 34.40/22.20 | 62.22/53.33 | **63.33/58.89** | 62.22/50.00 | 13.44/14.44 | 53.93/54.55 |
| Table | Vis (Pass@1) | 8.00/4.00 | 20.00/**36.00** | 20.00/28.00 | **24.00**/20.00 | 0.00/8.00 | 8.00/24.00 |

# F. Complete Validation Set Results

While the main text reports performance on the held-out **Test Set** (Table 2), we provide the corresponding **Validation Set** performance in Table 11.

**Strict Blind-Test Protocol.** To ensure rigorous evaluation, we enforced a strict separation of data splits during the agent's autonomous operation. The **Validation Set** was fully accessible to the agent, serving as the internal feedback mechanism for diagnosing failure modes, selecting optimal checkpoints, and refining data strategies during iterative loops. In contrast, the **Test Set** remained strictly invisible to the agent throughout the entire optimization process. All autonomous decisions—including when to stop training and which model to deploy—were based solely on validation metrics, ensuring that the reported test set performance reflects true generalization capability rather than overfitting to the evaluation metric.

*Table 11.* **Complete validation set results.** We report validation performance across 13 tasks in 5 domains. FT-Agent results are reported as mean±std over 3 independent runs. **Bold** indicates the best performance. ↑: higher is better; ↓: lower is better. *Abbreviations:* DA/FC/NR = Data Analysis/Fact Checking/Num. Reasoning; P@1 = Pass@1; MAE = Mean Absolute Error (for FG/Ring Count); TMS = Tanimoto Similarity; SR = Success Rate; VS = Valid SMILES; FTS = Fingerprint Similarity.

| | Math | Patent Examination | | | Table QA | | | | Finance | Chemistry | | | | | | | | |
| | AIME | PAR4PC | NOC4PC | PI4PC | DA | FC | NR | Vis. | QA | | Mol_Und | | Mol_Edit | Mol_Opt | | Reaction | |
| Method | Acc↑ | F1↑ | F1↑ | Acc↑ | Acc↑ | Acc↑ | Acc↑ | P@1↑ | Acc↑ | MAE↓ | TMS↑ | Acc↑ | Acc↑ | SR↑ | VS↑ | FTS↑ | Acc↑ |
|---|---|---|---|---|---|---|---|---|---|---|---|---|---|---|---|---|---|
| Qwen2.5-7B-Instruct | 0 | 60.63 | 27.40 | 29.00 | 28.87 | **83.30** | 40.00 | 8.00 | 73.00 | 0.52 | 0.09 | 56.00 | 34.40 | 21.00 | 66.00 | 14.93 | 36.00 |
| Manual SFT | 0 | **68.00** | 27.11 | 22.00 | **34.14** | 72.92 | 45.00 | **20.00** | 73.14 | 0.44 | 0.00 | 46.00 | 2.23 | 12.67 | 31.67 | 0 | 10.00 |
| OpenHands | 0 | 50.93 | 27.47 | **47.00** | 33.78 | 79.17 | 45.00 | 8.00 | **74.40** | 0.54 | 0.08 | 59.00 | 55.56 | 34.67 | 81.67 | 29.97 | **64.00** |
| FT-Agent | 13.32±6.67 | 67.50±6.89 | **34.44**±4.82 | 38.67±2.31 | 33.56±3.16 | 82.64±4.82 | **46.67**±4.62 | 16.00±4.00 | 74.04±1.10 | **0.24**±0.17 | **0.26**±0.08 | **62.78**±5.96 | **57.04**±5.01 | **36.22**±2.36 | **88.78**±6.38 | **30.64**±11.93 | 39.33±3.06 |

## G. Prompt Architecture

This section presents the prompt architecture used by FT-Agent, organized according to the four-phase workflow described in §3.2. Rather than relying on a single monolithic instruction, the framework implements a multi-stage prompting pipeline that mirrors how human researchers conduct benchmark analysis, model development, and experimental validation.

### G.1. Phase 1: Strategy Proposal

The strategy proposal phase generates unified hypotheses that jointly cover data processing and training configuration. To avoid repeating mistakes, the agent maintains a memory of historical experiments, inherits the current best configuration as baseline, and reviews sibling attempts before proposing changes.

---

**Unified Hypothesis Generation**

Generate a comprehensive hypothesis covering BOTH data processing AND training configuration.

**Data Processing**: Prioritize quality over quantity; apply code-based sampling within budget; use strong models for CoT generation with outcome-based validation.

**Training Configuration**: Select method based on sequence length and hardware constraints; balance overfitting risk against model quality based on dataset size.

**Output**: Structured JSON with reason, hypothesis, and implementation task.

---

### G.2. Phase 2: Implementation

The implementation phase translates hypotheses into executable code, comprising data processing scripts and training configurations. Both support a debug mode for rapid validation before full execution.

---

**Data Processing**

Generate a Python script to process the dataset according to the hypothesis.

**Execution**: Support debug mode (small subset) and full mode (within sample budget).

**Sampling**: Apply quality-first or diversity-preserving strategies based on hypothesis.

**CoT Requirement**: All training data must include step-by-step reasoning; validate that reasoning leads to correct answers.

**Output**: Alpaca-format JSON (instruction/input/output fields).

---

**Training Configuration**

Generate a training configuration based on the hypothesis and hardware constraints.

**Method Selection**: Choose based on required sequence length, dataset size, and hardware memory constraints.

---

**Batch Size**: Balance sequence length against batch size; use gradient accumulation for effective batch size.

**Validation**: Create validation split; align evaluation and save strategies.

### G.3. Phase 3: Fail-Fast Validation

The fail-fast validation phase implements progressive verification that catches errors early before full training runs. This corresponds to Stage 2 in the main text (§3.2).

---

**Data Validation**

Validate data processing output through three levels: (1) **Schema**: Output file exists and follows expected format. (2) **Content**: Required fields present in each sample. (3) **Quality**: CoT reasoning exists and answers follow benchmark format.

**Hard failures** (empty output, zero samples, missing CoT) require code fixes; **soft warnings** (high filter rate, skewed distribution) are logged but allow continuation.

---

**Config Validation**

Validate configuration through progressive stages: (1) **Syntax**: Configuration parses correctly. (2) **Compatibility**: Parameters valid for method and hardware. (3) **Execution**: Mini-batch training completes without error.

On failure, provide error classification, root cause analysis, and suggested fixes.

---

### G.4. Phase 4: Feedback Analysis

The feedback analysis phase evaluates completed experiments and generates actionable insights for the next iteration. This corresponds to Stage 3 (Structured Feedback Aggregation) in the main text.

---

**Experiment Feedback**

Analyze experiment results and decide whether to accept as new SOTA.

**Decision**: Accept if benchmark results exceed SOTA/baseline on primary metrics; reject otherwise.

**Multi-level Analysis**: (1) **Metrics**: Compare overall performance against baseline/SOTA. (2) **Error Patterns**: Identify capability gaps from failed samples (avoid benchmark-specific suggestions). (3) **Training Dynamics**: Analyze loss curves for overfitting/underfitting signs.

**Output**: Code summary, decision rationale with improvement suggestions, and accept/reject decision.

---

