# OpenReview forum: "FT-Dojo: Towards Autonomous LLM Fine-Tuning with Language Agents"
_ICML.cc/2026/Conference — ICML 2026 regular_

### Official Review · Reviewer_L4Af · 2026-03-10

**Soundness:** 3
**Presentation:** 3
**Significance:** 3
**Originality:** 3
**Overall Recommendation:** 4
**Confidence:** 3

**Summary:**

The paper introduces a benchmark and a system that evaluates and builds an autonomous system that mirrors human experts by leveraging evaluation-driven feedback to diagnose failures iteratively. The authors empirically demonstrate that it outperforms agents such as OpenHands.

The main contribution is to formalize LLM tuning in an interactive environment.

**Compliance With Llm Reviewing Policy:**

Affirmed.

**Final Justification:**

The authors addressed most of my concerns, therefore I am maintaining my rating.

**Key Questions For Authors:**

Please see above for questions.

Additional ones:

1. What rubrics and eval criteria were used for LLM-as-a-judge tasks. Prompt templates seem to be missing from the appendix.

**Limitations:**

Yes.

**Strengths And Weaknesses:**

Stength:

1. Timely benchmark to evaluate how well an agent performs in an open environment for a task that many people care about.

Weaknesses:

1. The authors should make it clearer what factors are controlled in the experiments. For instance, are tool accesses, agent scaffolding, and agent architectures controlled, and each experiment only varies one of them? Otherwise, it's not clear which parts of the agentic framework provide the desired improvement.

2. The authors downsample on many datasets, which is understandable given the resource constraints. However, it's not entirely clear on how this was done. Also, for some tasks, it might be worthwhile to further downsample, but repeat the experiments a few times to obtain a confidence interval, as agentic tasks are known to bear large variance.

3. If I understand correctly, the tasks in the benchmarks are not designed by the authors but aggregated from existing benchmarks, which makes claiming this as a benchmark slightly questionable.

---

> ### Author Rebuttal · Authors · 2026-03-30
>
> We thank the reviewer for the detailed and constructive feedback. We address each concern below.
>
> > Q1: The experimental controls are unclear; it is not clear which factors are varied across systems (e.g., tool access, scaffolding, architecture).
>
> We agree that the control protocol should be stated more explicitly. Across systems, we **hold fixed the external resources and runtime conditions**: the same base model, raw data repository, Docker sandbox, training/evaluation wrappers, LLM API access, and 12-hour wall-clock budget. What varies is the agent framework itself: FT-Agent introduces fine-tuning-oriented orchestration (structured iteration planning, fail-fast validation, and structured feedback analysis), while OpenHands retains a general-purpose CodeAct-style control policy under the same external tools. Our intention is to isolate the effect of fine-tuning-oriented agent design, rather than differences in tool availability or runtime conditions. We will revise the paper to make this factorization explicit.
>
> > Q2: Downsampling methodology unclear; need confidence interval
>
> We agree that the current paper under-specifies the held-out split construction, training data budget, and uncertainty reporting. We clarify each in turn.
>
> **Held-out split construction**: After context-length filtering, let n denote the number of remaining examples in each benchmark (or subtask). These are partitioned into non-overlapping validation and test sets according to:
>
> $$N_{val} = N_{test} = \min(100, \lfloor n/2 \rfloor),$$
>
> All agent decisions are based solely on validation feedback; the test set remains inaccessible throughout optimization, ensuring strict separation between the two splits.
>
> **Training data budget**: The 2,000-example limit is **enforced as a shared resource budget rather than a fixed precomputed subset**. Under this common cap, each method independently selects or constructs its own training data according to its own procedure. We agree the current wording may suggest a single global downsampling rule, and will revise accordingly.
>
> Uncertainty reporting: We have conducted 3 independent runs of FT-Agent and now report mean ± std across all tasks. **Please see our response to Reviewer WDYu, Q2 for the full results and variance analysis.**
>
> > Q3: Tasks aggregated from existing benchmarks—is this truly a benchmark?
>
> We appreciate this important conceptual question. We agree that FT-Dojo is not a collection of entirely newly created static datasets, and we will revise the framing to make the contribution more precise.
>
> The object benchmarked by FT-Dojo is not standalone model accuracy on raw test sets, but the end-to-end autonomous fine-tuning process under a unified interactive protocol. Concretely, FT-Dojo standardizes: (1) the task interface, (2) the open-ended data repository from which training data must be constructed, (3) the sandboxed execution environment, (4) the validation/test evaluation protocol, and (5) the structured feedback returned after each iteration. Existing benchmarks are used as held-out evaluators for domain-specific target capabilities, but the benchmarked object is the agent's ability to iteratively build data, configure training, run fine-tuning, and improve from feedback.
>
> This design follows an established pattern: SWE-Bench aggregates GitHub issues into an interactive coding benchmark, and MLE-Bench repurposes Kaggle competitions to evaluate ML engineering agents. In both cases, the novelty lies in the evaluation protocol and interactive scaffold, not the underlying data. FT-Dojo follows the same principle—it is best understood as an interactive benchmark environment for autonomous LLM fine-tuning, rather than a claim that all constituent datasets are newly introduced. We will revise the paper to make this framing explicit.
>
> > Q4: What rubrics and eval criteria were used for LLM-as-a-judge tasks. Prompt templates seem to be missing from the appendix.
>
> We thank the reviewer for catching this omission. Most tasks use rule-based or task-specific evaluators (exact match, accuracy, or domain-specific custom evaluators). The main LLM-judge case is AIME 2025, where we use a cascade evaluator: a rule-based math verifier is applied first, and only unresolved cases are sent to a fixed GPT-5 judge with an identical grading prompt across all methods.
>
> The judge receives the original question, gold answer, and model prediction, and assesses correctness only. The rubric allows mathematically equivalent expressions, requires all parts correct for multi-part answers, and ignores formatting differences such as \boxed{} markers. The judge outputs a binary decision (CORRECT / INCORRECT). We will add the exact prompt template and rubric to the appendix and clarify which tasks use LLM-as-a-judge versus deterministic evaluators.

---

> > ### Author Rebuttal · Reviewer_L4Af · 2026-03-31
> >
> > The authors addressed most of my concerns; I understand that some could not be fleshed out due to space constraints.

---

> > > ### Author Response · Authors · 2026-04-04
> > >
> > > Thank you sincerely for your thoughtful and constructive feedback. We apologize that space constraints in the rebuttal prevented us from elaborating on all points as thoroughly as they deserved. **We want to assure you that all the clarifications discussed—including the experimental control protocol, downsampling and split construction methodology, benchmark framing, and the LLM-as-a-judge prompt templates and rubrics—will be carefully and fully incorporated into the final version of the manuscript**. We also hope the additional frontier-agent baselines (Codex and Claude Code) and open-source backbone experiments (DeepSeek-V3.2 and Qwen3.5-397B-A17B) presented in our response to Reviewer bVxX, Q2 may serve as a useful reference, as they further substantiate the robustness and generality of our evaluation.
> > >
> > > We truly appreciate your support and will thoughtfully incorporate all suggested revisions.
> > >
> > > **Best regards,**
> > >
> > > **Authors**

---

### Official Review · Reviewer_bVxX · 2026-03-10

**Soundness:** 3
**Presentation:** 4
**Significance:** 3
**Originality:** 3
**Overall Recommendation:** 5
**Confidence:** 2

**Summary:**

This paper introduces FT-Dojo, a novel interactive environment designed to evaluate the capability of Large Language Models (LLMs) to function as autonomous agents that perform end-to-end fine-tuning. Spanning 13 tasks across 5 domains, FT-Dojo tests an agent's ability to curate data, configure training hyperparameters, and iteratively improve models based on structured evaluation feedback. A major issue addressed by the study is that general-purpose agents (such as OpenHands) consistently fail in this specialized domain because they get overwhelmed by verbose logs, waste compute on inefficient exploration, and struggle to interpret complex, multifaceted training feedback. To solve this, the authors propose FT-Agent, an autonomous framework equipped with structured iteration planning, fail-fast validation, and structured feedback analysis. Overall, a central theme considered by the study is evaluating LLMs at the algorithmic level in an open-ended optimization landscape, pushing beyond standard machine learning engineering benchmarks. FT-Agent sets a new baseline, outperforming OpenHands on 10 out of 13 tasks.

**Compliance With Llm Reviewing Policy:**

Affirmed.

**Final Justification:**

I thank the authors for their efforts. I'll maintain my rating as 5.

**Key Questions For Authors:**

I am curious about the following questions:
[I] How does FT-Agent perform when its interactive planning backbone is entirely driven by state-of-the-art open-weight models (such as Llama-3-70B) rather than GPT-5.2 or GPT-4o? I understand that the authors acknowledge the need for frontier intelligence, but it should be good to include an open weight counterpart that similarly benchmarks to GPT-5.2.
[II] If the 12-hour compute budget is relaxed to 24 or 48 hours, do general-purpose agents like OpenHands eventually converge on effective fine-tuning configurations, or do their causal reasoning deficits prevent them from succeeding regardless of time?

**Limitations:**

Yes

**Strengths And Weaknesses:**

Strengths:
[I] The Benchmark Itself: By formalizing the end-to-end LLM fine-tuning process as a structured agent task, the authors have created a timely and highly relevant benchmark. FT-Dojo's inclusion of data strategies alongside hyperparameter tuning as first-class optimization targets accurately reflects real-world ML engineering.
[II] The "fail-fast validation" methodology seems like a highly effective architectural decision. By catching syntax errors, incompatible formats, and exploding losses via mini-runs, FT-Agent achieves significantly higher iteration throughput against the OpenHands baseline within the same budget.
[III] The paper presents excellent qualitative case studies that contrast cumulative learning with shotgun debugging. Identifying that current agents lack causal reasoning and often fall back on indiscriminate trial-and-error when facing complex training failures highlights the boundaries of today's autonomous ML.

Weakness:
[I] I am curious about what the time budget indicates, as the 12-hour time budget can favor the "fail-fast" approaches, and whether generalist baselines like OpenHands inherently cannot solve these tasks.

---

> ### Author Rebuttal · Authors · 2026-03-30
>
> We thank the reviewer for the positive assessment and constructive questions. We address the weakness and key questions below.
>
> > Q1: Does the 12-hour time budget inherently favor fail-fast approaches, and would OpenHands perform better given more time?
>
> We agree that this is an important concern, since a fixed wall-clock budget could in principle favor fail-fast strategies. To investigate this directly, we ran extended experiments on AIME 2025—one of the most challenging tasks—by doubling OpenHands' wall-clock budget from 12 to 24 hours. **Across 2 runs, we observed no meaningful performance improvement over the 12-hour setting**. In both runs, the best-performing checkpoint appeared within the first 6 hours, and additional iterations beyond that point did not lead to further gains.
>
> This is consistent with the data bottleneck in the raw AIME training repository provided in FT-Dojo. As shown in Table 1, 82% of the raw training samples available to the agent lack solutions and contain no chain-of-thought rationale, making the provided data sparse, noisy, and largely ineffective for SFT without active synthesis. Under these conditions, simply allocating more time does not help if the agent cannot identify and address the true failure mode. OpenHands encounters its primary difficulty at this stage: it treats data processing as generic script execution and lacks the structured feedback analysis needed to diagnose that missing CoT is the key bottleneck. This is not due to a lack of tool access: OpenHands is equipped with the **same LLM API endpoints for data synthesis as FT-Agent**. The gap arises from its inability to formulate the right hypothesis—recognizing that CoT synthesis is needed—rather than from a lack of the capability to execute it.
>
> We therefore view the benefit of fail-fast design not as exploiting a particular time constraint, but as reallocating limited compute toward higher-value exploration. A larger budget may reduce the severity of this issue, but does not remove the underlying need for structured diagnosis and early validation.
>
> > Q2: How does FT-Agent perform with open-weight frontier backbone models?
>
> We address this question jointly with the missing frontier-agent baseline comparison raised by **Reviewer JWwB & WDYu**, as both share an identical task setup and together offer a comprehensive view of FT-Agent's generality. All experiments use the same 5 representative tasks from §4.3 under strictly controlled conditions: the same LlamaFactory/OpenCompass wrappers, Docker environment, data repository, and 12-hour wall-clock budget. Results are summarized in the table below.
>
> | Method | Backbone | AIME | PI4PC | Mol_Edit | Finance QA | Vis. | Avg |
> |-|-|-|-|-|-|-|-|
> | Base Model | — | 0.00 | 34.00 | 22.20 | 65.10 | 4.00 | 25.06 |
> | FT-Agent | Qwen3.5-397B-A17B (Open-source) | 6.67 | 24.00 | 34.00 | 67.16 | 28.00 | 31.97 |
> | OpenHands | GPT-5.2 | 0.00 | 44.00 | 40.00 | 64.73 | 24.00 | 34.55 |
> | FT-Agent | DeepSeek-V3.2 (Open-source) | 6.67 | 45.00 | 50.00 | 65.47 | 12.00 | 35.83 |
> | Codex | GPT-5.2 | 6.67 | 42.00 | 40.00 | 65.63 | 26.00 | 36.06 |
> | FT-Agent | GPT-4o | **13.33** | 29.00 | 50.00 | **70.95** | 20.00 | 36.65 |
> | Claude Code | Claude Sonnet-4.6-thinking | 0.00 | **51.00** | 44.00 | 68.32 | **36.00** | 39.86 |
> | FT-Agent | GPT-5.2 | **13.33** | 44.00 | **53.33** | 67.53 | **36.00** | **42.83** |
>
> First, FT-Agent (GPT-5.2) achieves **the highest average score across all systems**, outperforming Claude Code despite the latter using a strictly stronger backbone (Sonnet-4.6-thinking). The gap is most pronounced on AIME, the most challenging task where the majority of training samples lack ground-truth solutions and CoT annotations: FT-Agent scores 13.33% while Claude Code scores 0%. This suggests that raw model capability alone is not the primary bottleneck—the ability to diagnose data quality issues, synthesize CoT annotations from noisy signals, and act on multi-level feedback is what differentiates systems on hard tasks.
>
> Second, among GPT-5.2-based agents, FT-Agent outperforms both OpenHands and Codex by a clear margin despite sharing the same backbone. Notably, Codex—OpenAI's **natively integrated framework**—achieves a modest improvement over OpenHands, confirming that the bottleneck lies not in code generation quality but in data quality diagnosis and multi-level feedback interpretation, capabilities that FT-Agent is specifically designed to address.
>
> Third, open-source backbones achieve competitive results within FT-Agent's framework, demonstrating that **performance gains are framework-driven rather than backbone-dependent**. DeepSeek-V3.2 **surpasses OpenHands and matches Codex** without any proprietary model access. Qwen3.5-397B-A17B shows a lower overall average, yet on AIME both open-source variants still match Codex and outperform Claude Code (Sonnet-4.6-thinking). This further reinforces that FT-Agent's gains are framework-driven and generalize across backbone types.

---

> > ### Author Rebuttal · Reviewer_bVxX · 2026-04-01
> >
> > Thank you for addressing my questions; my concerns have been adequately addressed.

---

> > > ### Author Response · Authors · 2026-04-04
> > >
> > > Thank you sincerely for your thoughtful and constructive feedback. We are glad that the extended-budget experiments and the open-source backbone results (DeepSeek-V3.2 and Qwen3.5-397B-A17B) were helpful in addressing your questions. We will incorporate these additional results and discussions into the final version of the manuscript.
> > >
> > > **Best regards,**
> > >
> > > **Authors**

---

### Official Review · Reviewer_WDYu · 2026-03-12

**Soundness:** 2
**Presentation:** 3
**Significance:** 3
**Originality:** 3
**Overall Recommendation:** 5
**Confidence:** 4

**Summary:**

This paper proposes FT-Dojo, an interactive benchmark with 13 tasks across 5 domains for evaluating autonomous end-to-end LLM fine-tuning, where agents must independently navigate data curation, training configuration, and iterative refinement from a shared heterogeneous data repository. It also introduces FT-Agent, which employs structured iteration planning, fail-fast validation, and multi-level feedback analysis to address context explosion, inefficient exploration, and poor feedback interpretation in fine-tuning scenarios. FT-Agent achieves the best performance on 10 of 13 tasks, significantly outperforming the OpenHands baseline.

**Compliance With Llm Reviewing Policy:**

Affirmed.

**Final Justification:**

FT-Dojo is a well-motivated benchmark with broad task coverage, and FT-Agent's three-stage design is principled and clearly presented. My main concern was the fairness of the Manual SFT baseline due to the iteration count disparity (2 vs. 8.77). The rebuttal partially addressed this by providing "Manual SFT w/ LLM synthesis" results, though the iteration imbalance remains uncontrolled. The additional multi-run statistics and stronger baselines (Codex, Claude Code) further strengthened the evaluation. I am raising my score to 5, provided the final version explicitly acknowledges the limitations of the Manual SFT baseline and avoids overstating comparisons to human expert capability.

**Key Questions For Authors:**

See weakness.

**Limitations:**

yes

**Strengths And Weaknesses:**

Strengths:
1. The paper focuses on data cleaning and iterative fine-tuning for deploying LLMs in vertical domains, which is a meaningful research topic.
2. FT-Dojo offers broad task coverage, spanning multiple tasks across five domains.
3. FT-Agent has a clear design logic, with its three modules neatly corresponding to the three identified challenges.

Weaknesses:
1. Fairness of the Manual SFT baseline: Section 4.2 notes that much of the data lacks complete CoT, resulting in 0% accuracy for the manual expert. However, FT-Agent can improve performance by synthesizing data. For a fair comparison, the manual expert should also be allowed to call LLMs for data synthesis, just as FT-Agent does. Additionally, the manual expert is restricted to only two iterations, which is a significant disadvantage compared to FT-Agent's average of 8.77 loops.
2. Evaluation robustness: Agents inherently exhibit considerable randomness, so how much variance would arise from multiple runs on the same task? Moreover, many test sets contain fewer than 100 samples. As a benchmark, results should be reported over multiple runs.
3. Insufficient baselines: The paper only compares FT-Agent against OpenHands, which is limited. Including other agentic pipelines would provide a more comprehensive evaluation.

---

> ### Author Rebuttal · Authors · 2026-03-30
>
> We thank the reviewer for the constructive feedback and positive recognition of the benchmark coverage and FT-Agent's design logic. We address each concern below.
>
> > Q1: The Manual SFT baseline is not fair, because it is more constrained than FT-Agent.
>
> We acknowledge this concern and agree that the original Manual SFT baseline should be defined more carefully. More precisely, it is intended as **a constrained, resource-matched human baseline**, rather than an expert upper bound or a fully assisted human best-effort baseline.
>
> In the original setup, practitioners followed a standard manual workflow **without LLM assistance** (Appendix B): rule-based data cleaning, manual hyperparameter tuning, and evaluation-based refinement, restricted to a two-pass protocol within the same 12-hour budget. The weak results on tasks such as AIME and Mol_Edit reflect both the constrained protocol and the inherent difficulty of these tasks under rule-based-only processing.
>
> To address this concern, we additionally ran Manual SFT with explicit LLM access for CoT synthesis and data cleaning:
>
> | Method | AIME | PI4PC | Vis. | Finance QA | Mol_Edit |
> |-|-|-|-|-|-|
> | Manual SFT (original) | 0.00 | 28.00 | 8.00 | 71.68 | 0.00 |
> | Manual SFT (w/ LLM synthesis) | 20.00 | 28.00 | 20.00 | 67.57 | 50.00 |
> | FT-Agent | 13.33 | 44.00 | 36.00 | 67.53 | 53.33 |
>
> With LLM assistance and a relaxed iteration protocol, the human baseline improves substantially on tasks where the original constrained protocol produced near-zero results (AIME, Vis., Mol_Edit). FT-Agent achieves competitive performance across all evaluated tasks while operating **fully autonomously without any human guidance or manual iteration**.
>
> We will revise the paper to narrow the associated claims: FT-Agent automates these capabilities end-to-end without human involvement, and is best understood as a strong initial baseline for future research rather than a definitive solution.
>
> > Q2: The evaluation is not sufficiently robust, since agentic results are stochastic and many test sets are small; benchmark results should be reported over multiple runs.
>
> We thank the reviewer for this important point and agree that multi-run reporting is important for benchmark-quality evaluation of stochastic agentic systems. To address this concern, we additionally report mean ± std over 3 independent runs for FT-Agent below:
>
> | Task | Original (single run) | Updated (mean±std, 3 runs) |
> |---|---|---|
> | AIME ↑ | 13.33 | 11.11±3.82 |
> | PAR4PC ↑ | 70.34 | 67.61±2.40 |
> | NOC4PC ↑ | 41.09 | 36.81±4.13 |
> | PI4PC ↑ | 44.00 | 49.00±4.36 |
> | DA ↑ | 34.48 | 34.71±0.28 |
> | FC ↑ | 83.33 | 81.25±2.08 |
> | NR ↑ | 46.00 | 45.33±1.15 |
> | Vis. ↑ | 36.00 | 29.33±6.11 |
> | Finance QA ↑ | 67.53 | 67.20±0.31 |
> | Mol_Und (MAE) ↓ | 0.42 | 0.23±0.16 |
> | Mol_Und (TMS) ↑ | 0.42 | 0.36±0.06 |
> | Mol_Und (Acc) ↑ | 61.67 | 67.55±5.68 |
> | Mol_Edit ↑ | 53.33 | 54.44±2.94 |
> | Mol_Opt (SR) ↑ | 36.67 | 34.44±2.17 |
> | Mol_Opt (VS) ↑ | 91.67 | 90.00±4.41 |
> | Reaction (FTS) ↑ | 18.11 | 30.92±11.22 |
> | Reaction (Acc) ↑ | 38.00 | 35.33±3.06 |
>
> The results confirm that the main qualitative conclusions hold across runs. Reaction (FTS) exhibits higher variance (±11.22), which we attribute to the sensitivity of molecular fingerprint similarity to subtle differences in generated SMILES strings across runs.
>
> We note that 3 seeds × 4 methods × 13 tasks would require over 1,500 GPU-hours, making exhaustive multi-run evaluation prohibitive; we therefore report multi-run results for FT-Agent only. Nevertheless, the key behavioral differences between FT-Agent and OpenHands are **structural rather than stochastic**: the iteration count gap (8.77 vs 3.69) reflects a deterministic property of each agent's control flow—how it allocates its budget between validation and exploration—which is largely independent of random seed.
>
> > Q3: The baseline comparison is limited, since FT-Agent is only compared against OpenHands.
>
> We agree that the original baseline set was limited. In response, we have added **Codex** and **Claude Code** as frontier-agent baselines, and additionally evaluated FT-Agent with **two open-source frontier backbones** (DeepSeek-V3.2 and Qwen3.5-397B-A17B) to provide a more comprehensive view of the framework's generality. All experiments are conducted on the same representative tasks used in our ablation study **under identical conditions**: same LlamaFactory/OpenCompass wrappers, Docker environment, data repository, and 12-hour wall-clock budget. Across all systems, FT-Agent (GPT-5.2) achieves **the highest average score**—outperforming Codex, OpenAI's **natively integrated** coding framework sharing the same backbone, and further surpassing Claude Code despite the latter using a stronger backbone. Open-source variants further confirm that these gains are framework-driven rather than backbone-dependent. **Please see our response to Reviewer bVxX, Q2 for the full results table and extended analysis.**

---

> > ### Author Rebuttal · Reviewer_WDYu · 2026-04-03
> >
> > Thank you for the additional experiments. Regarding Q1, I appreciate the new "Manual SFT w/ LLM synthesis" results, which partially address my concern. However, since the iteration count disparity (2 vs. 8.77 on average) remains uncontrolled at the experimental level, I recommend that the authors tone down the claims around the Manual SFT baseline in the final version, explicitly acknowledging its limitations (i.e., the restricted number of iterations), and avoid presenting it as representative of human expert capability. With this revision, I am willing to raise my score to 5.

---

> > > ### Author Response · Authors · 2026-04-04
> > >
> > > Thank you sincerely for your thorough and constructive engagement throughout the review process. We fully acknowledge your remaining concern regarding the iteration count disparity between the Manual SFT baseline and FT-Agent. In the final version, we will **explicitly** note the constrained nature of the original protocol and ensure that the associated claims are appropriately scoped, refraining from presenting the Manual SFT baseline as representative of human expert capability. We look forward to incorporating all suggested revisions into the manuscript.
> > >
> > > **Best regards,**
> > >
> > > **Authors**

---

### Official Review · Reviewer_JWwB · 2026-03-12

**Soundness:** 3
**Presentation:** 2
**Significance:** 4
**Originality:** 4
**Overall Recommendation:** 4
**Confidence:** 5

**Summary:**

This paper studies a timely and important problem: whether language agents can autonomously perform end-to-end LLM fine-tuning, including data curation/construction, training configuration, and iterative improvement based on evaluator feedback. I think the paper makes a meaningful initial step toward formalizing this problem, and FT-Dojo is potentially a valuable benchmark/infrastructure contribution for the community.

However, I do not think the current evaluation setup is strong enough to support the paper’s stronger methodological claims. In particular, the baseline set is too weak, the role of the agent framework vs the role of fine-tuning-specific design is not clearly disentangled, and FT-Agent itself is not yet convincingly established as a generally meaningful method rather than a benchmark-specific engineered system.

Overall, I view this paper as a promising benchmark or problem-definition paper, but not yet a convincing demonstration that the proposed FT-Agent represents a substantial methodological advance.

**Compliance With Llm Reviewing Policy:**

Affirmed.

**Final Justification:**

The author have addressed all my concern, so i have raised my score to 4.

**Key Questions For Authors:**

1. Can the authors evaluate against substantially stronger agent baselines, especially more capable frontier-style coding agents, rather than mainly OpenHands?

2. Can the authors provide a much more rigorous Manual SFT baseline, with clearer protocol details and stronger evidence that it is actually competitive?

3. How do the authors envision scaling this benchmark beyond the current task set? In my view, the long-term impact of this work depends heavily on whether it can support scalable data construction and evaluation on stronger models and stronger agent systems.

**Limitations:**

yes

**Strengths And Weaknesses:**

##  Strengths
The problem is well motivated and comes at the right time. As agent systems are increasingly used for coding, experimentation, and research workflows, it is natural and important to ask whether they can also automate fine-tuning workflows rather than only inference-time tool use.

The benchmark contribution is the strongest part of the paper. FT-Dojo gives the community an initial interactive environment in which autonomous fine-tuning can be evaluated in a more systematic way. I think this aspect has real potential to matter, and could become the basis for future work on agentic post-training or agent-for-research settings.

The paper also identifies several real difficulties of this setting, including long iterative context, expensive fail-slow exploration, and the need to interpret multi-level training feedback. These are real issues, and the paper is directionally correct in arguing that this problem is not captured well by standard tool-use benchmarks.


##  Weaknesses

My main concern is that the evaluation is not well matched to the ambition of the claims.

First, the baselines are too weak. The comparisons are mainly against the base model, Manual SFT, and OpenHands. This is not sufficient for establishing that FT-Agent is a meaningful methodological advance. In particular, the paper does not compare against sufficiently strong modern agent systems (codex, claude code). As a result, the experiments currently show that a benchmark-specific specialized system can outperform a relatively weak comparison set, but they do not convincingly show that FT-Agent captures something fundamental about autonomous fine-tuning.

Second, the OpenHands comparison is conceptually blurry. The paper does not cleanly separate:

1. gains from using an agent framework
2. gains from using an agent to run a fine-tuning loop

This makes the empirical conclusions hard to interpret. A reader is left unsure whether FT-Agent is truly better because of its fine-tuning-specific design, or simply because it is more heavily engineered for this benchmark than the compared general-purpose agent baseline.

Third, the Manual SFT baseline is not convincing. Several results are unexpectedly weak, including near-zero outcomes on some tasks. That does not strengthen the paper’s claims; instead it raises concern that the baseline is underpowered or insufficiently documented. If the paper wants to argue that the agent is competitive with or better than expert manual fine-tuning, it needs a much stronger and more transparent human baseline.

Finally, the current setup feels too tied to relatively small-model fine-tuning under a constrained budget. If the authors want this line of work to become broadly influential, and especially relevant to frontier LLM development, the benchmark needs to evolve toward stronger model settings, stronger agent frameworks, and scalable data sourcing/collection mechanisms. Otherwise, the work risks remaining a narrow benchmark exercise rather than becoming a task abstraction that frontier labs would actually adopt.

---

> ### Author Rebuttal · Authors · 2026-03-30
>
> We thank the reviewer for the thorough evaluation and for recognizing the benchmark's potential and the problem's timeliness. We address each concern below.
>
> > Q1: Missing strong frontier-agent baselines.
>
> We agree that the original baseline set was limited. In response, we have added **Codex (GPT-5.2)** and **Claude Code (Claude Sonnet-4.6-thinking)** as frontier-agent baselines, and additionally evaluated FT-Agent with **two open-source frontier backbones** (DeepSeek-V3.2 and Qwen3.5-397B-A17B) to provide a more comprehensive view of the framework's generality. All experiments are conducted on the same representative tasks used in our ablation study **under identical conditions**: same LlamaFactory/OpenCompass wrappers, Docker environment, data repository, and 12-hour wall-clock budget. Across all systems, FT-Agent (GPT-5.2) achieves **the highest average score**—outperforming Codex, OpenAI's **natively integrated** coding framework sharing the same backbone, and further surpassing Claude Code despite the latter using a stronger backbone. Open-source variants further confirm that these gains are framework-driven rather than backbone-dependent. **Please see our response to Reviewer bVxX, Q2 for the full results table and extended analysis.**
>
> > Q2: The OpenHands comparison does not cleanly disentangle fine-tuning-oriented design from benchmark-specific engineering.
>
> We understand this concern and agree that the paper should state this distinction more clearly. The key question is whether FT-Agent's gains come from fine-tuning-oriented design or simply from heavier benchmark-specific engineering.
>
> The quantitative evidence supports the former. As shown in Table 3, FT-Agent averages 8.77 iterations per task versus 3.69 for OpenHands, while achieving a higher improvement rate. This gap is driven by fail-fast validation and structured feedback analysis—not by additional tool access or benchmark-specific shortcuts. The frontier-agent results further reinforce this: **Claude Code, with a stronger backbone and strong coding capabilities, still cannot match FT-Agent on AIME under identical conditions**, making it unlikely that FT-Agent's advantage is simply more aggressive benchmark engineering.
>
> FT-Agent's three mechanisms are derived from real-world expert fine-tuning practices—managing verbose logs, validating before costly runs, and interpreting multi-level feedback—rather than reverse-engineered from FT-Dojo's interface. FT-Agent interacts with FT-Dojo through the same Meta API as all other agents. We do not claim FT-Agent as a universally superior agent framework, but rather position it as a strong initial baseline that demonstrates the value of fine-tuning-oriented design for future research on autonomous fine-tuning.
>
> > Q3: The Manual SFT baseline is unconvincing, with near-zero results on some tasks and insufficient documentation.
>
> The original Manual SFT baseline was designed as **a constrained, resource-matched human baseline without LLM assistance (Appendix B)**. To address this concern, we additionally ran an LLM-assisted Manual SFT, which substantially improved results on the most challenging tasks (e.g., AIME: 0→20%, Vis.: 8→20%, Mol_Edit: 0→50%). FT-Agent achieves comparable or superior performance fully autonomously on the majority of tasks. **Please see our response to Reviewer WDYu, Q1 for the full discussion and updated results.**
>
> > Q4: The current setting may be too tied to small-model, constrained-budget fine-tuning to support broader frontier relevance.
>
> We agree that stronger model settings, more capable agent frameworks, and scalable data sourcing are important future directions. The current work is best understood as an exploratory first-step benchmark: its goal is to formalize autonomous LLM fine-tuning as an agent task for the first time, not to claim coverage of frontier-scale scenarios.
>
> FT-Dojo is designed for extensibility: adding new tasks requires only a compatible data source, task objective, and evaluation script, with no changes to core infrastructure (Section 2.4). More importantly, the core challenges FT-Dojo addresses—data curation, training configuration, fail-fast validation, and iterative refinement from feedback—are not specific to small models; if anything, they become more pronounced at larger scale.
>
> We also note that the fixed data and compute conditions are **a deliberate design choice, not a limitation**. FT-Dojo evaluates an agent's ability to leverage a given data repository—the core challenge when data sourcing is already partially addressed. Allowing unconstrained external access would conflate fine-tuning ability with information retrieval, eliminating comparability. This mirrors standard benchmark practice of fixing the task context to isolate the target capability. Open-world data sourcing is a natural future direction, but constitutes a distinct task formulation. We will revise the paper to make this framing explicit.

---

> > ### Author Rebuttal · Reviewer_JWwB · 2026-03-31
> >
> > I have raised my score to 4. Good paper.

---

> > > ### Author Response · Authors · 2026-04-04
> > >
> > > Thank you once again for your kind and constructive feedback. We will revise the paper to more explicitly articulate the distinction between framework-driven gains and benchmark-specific engineering, and will include the frontier-agent baseline results (Codex and Claude Code) as well as the open-source frontier backbone experiments within the FT-Agent framework in the final version.
> > >
> > > We truly appreciate your support and will thoughtfully incorporate all suggested revisions.
> > >
> > > **Best regards,**
> > >
> > > **Authors**

---

### Decision · Program_Chairs · 2026-04-30

**Decision:**

Accept (regular)

**Comment:**

The paper presents a benchmark for evaluating autonomous LLM fine-tuning agents, covering 13 tasks across 5 domains, together with an agent system (FT-Agent) designed for this setting. The benchmark addresses a genuine gap — we lack systematic ways to evaluate agents that perform end-to-end fine-tuning — and the design is thorough. The rebuttal added comparisons with frontier agents (Codex, Claude Code), experiments on open-source backbones, and multi-run statistics, all of which strengthened the empirical story. FT-Agent held up across these settings. The reviewers are broadly positive, though several note that the paper is more of a benchmark contribution than a methodological one. I agree with this characterization, but well-designed benchmarks are valuable, and this one is timely. Recommended for acceptance as a benchmark paper.